# Impacts of Radiometric Uncertainty and Weather-Related Surface Conditions on Soil Moisture Retrievals with Sentinel-1

**Harm-Jan F. Benninga \*** , **Rogier van der Velde** and **Zhongbo Su**

Department of Water Resources, Faculty of Geo-Information Science and Earth Observation (ITC),
University of Twente, P.O. Box 217, 7500 AE Enschede, The Netherlands
\* Correspondence: h.f.benninga@utwente.nl

**Abstract:** The radiometric uncertainty of Synthetic Aperture Radar (SAR) observations and weather-related surface conditions caused by frozen conditions, snow and intercepted rain affect the backscatter ($\sigma^0$) observations and limit the accuracy of soil moisture retrievals. This study estimates Sentinel-1's radiometric uncertainty, identifies the effects of weather-related surface conditions on $\sigma^0$ and investigates their impact on soil moisture retrievals for various conditions regarding soil moisture, surface roughness and incidence angle. Masking rules for the surface conditions that disturb $\sigma^0$ were developed based on meteorological measurements and timeseries of Sentinel-1 observations collected over five forests, five meadows and five cultivated fields in the eastern part of the Netherlands. The Sentinel-1 $\sigma^0$ observations appear to be affected by frozen conditions below an air temperature of $1\,^\circ$C, snow during Sentinel-1's morning overpasses on meadows and cultivated fields and interception after more than $1.8\,$mm of rain in the $12\,$h preceding a Sentinel-1 overpass, whereas dew was not found to be of influence. After the application of these masking rules, the radiometric uncertainty was estimated by the standard deviation of the seasonal anomalies timeseries of the Sentinel-1 forest $\sigma^0$ observations. By spatially averaging the $\sigma^0$ observations, the Sentinel-1 radiometric uncertainty improves from $0.85\,$dB for a surface area of $0.25\,$ha to $0.30\,$dB for $10\,$ha for the VV polarization and from $0.89\,$dB to $0.36\,$dB for the VH polarization, following approximately an inverse square root dependency on the surface area over which the $\sigma^0$ observations are averaged. Deviations in $\sigma^0$ were combined with the $\sigma^0$ sensitivity to soil moisture as simulated with the Integral Equation Method (IEM) surface scattering model, which demonstrated that both the disturbing effects by the weather-related surface conditions (if not masked) and radiometric uncertainty have a significant impact on the soil moisture retrievals from Sentinel-1. The soil moisture retrieval uncertainty due to radiometric uncertainty ranges from $0.01\,$m$^3$ m$^{-3}$ up to $0.17\,$m$^3$ m$^{-3}$ for wet soils and small surface areas. The impacts on soil moisture retrievals are found to be weakly dependent on the surface roughness and the incidence angle, and strongly dependent on the surface area (or the $\sigma^0$ disturbance caused by a weather-related surface condition for a specific land cover type) and the soil moisture itself.

**Keywords:** Sentinel-1; radiometric uncertainty; disturbing surface conditions; masking rules; soil moisture

## 1. Introduction

Earth observations made by Synthetic Aperture Radar (SAR) instruments can be used to estimate soil moisture at field scale (e.g., [1–5]). However, SAR backscatter ($\sigma^0$) observations contain inaccuracies due to calibration uncertainties, sensor instabilities and speckle effects [6–9], hereafter combined and

referred to as radiometric uncertainty. In addition to imperfections inherent to a retrieval algorithm, the radiometric uncertainty controls the accuracy of soil moisture retrievals.

The radiometric uncertainty of SAR observations can be determined by analysing the temporal $\sigma^0$ variability of a target for which the $\sigma^0$ is time-invariant [10–12]. The C-band Sentinel-1 signal is unlikely to penetrate dense canopies [6,13–15]. Therefore, forest $\sigma^0$ can be assumed to be unaffected by variations in soil moisture and surface roughness, which can cause large and abrupt changes in the $\sigma^0$ over other land covers, and forests should behave as fairly time-invariant targets [10–12]. This property of the forest $\sigma^0$ was exploited to estimate the radiometric stability of SAR observations by calculating the standard deviation of the $\sigma^0$ timeseries [10–12].

Previous studies [10–12] determined the radiometric stability of $\sigma^0$ timeseries from targets with surface areas ranging from a pine tree forest of 8.1 ha (856 Sentinel-1 pixels) to an Amazon rainforest area of $10\,000\,\text{km}^2$. Calibration uncertainties and sensor instabilities will mainly determine the variability in the mean forest $\sigma^0$ over such large surface areas, because the large number of independent samples leads to the suppression of the speckle effects [6,12]. Radiometric uncertainty from smaller surface areas, which contains a larger contribution of speckle due to the lower number of independent samples, is particularly relevant for the retrieval of soil moisture at field scale. If speckle is not completely suppressed, this is a contributing factor to the soil moisture retrieval accuracy. For example, with synthetically generated speckled $\sigma^0$ representing Sentinel-1 observations of bare soil, Pierdicca et al. [16] obtained an improvement in soil moisture retrieval accuracy (root mean squared deviation, $E_{RMS}$) from $0.035\,\text{m}^3\,\text{m}^{-3}$ at the native resolution (1 pixel) to $0.025\,\text{m}^3\,\text{m}^{-3}$ at field scale ($10 \times 10$ pixels). For the quantification of radiometric uncertainty and its effect on soil moisture retrieval uncertainty, radiometric uncertainty needs to be determined as a function of the surface area over which the $\sigma^0$ observations are aggregated.

Besides radiometric uncertainty, several weather-related surface conditions may disturb $\sigma^0$ observations, e.g., inundation [17,18], frozen conditions [19–21], wet snow [6,22,23], intercepted rain [10,24–26] and dew [6,27–29], and thus affect the forthcoming soil moisture retrievals. Masking satellite observations for conditions under which soil moisture retrieval is not possible or more uncertain is common for existing coarse resolution ($> 9\,\text{km}$) products, such as ASCAT [30,31], SMOS [32,33] and SMAP [33]. Previous studies [34,35] have acknowledged that geophysical products derived from Sentinel-1 are also subject to elevated uncertainty levels if unwanted disturbances in the $\sigma^0$ observations are not addressed and further development of operational products would benefit from a masking procedure for disturbing surface conditions.

The objectives of this study are (1) to develop a masking procedure for the weather-related surface conditions that disturb Sentinel-1 $\sigma^0$ observations, (2) to estimate Sentinel-1's radiometric uncertainty as a function of surface area, and (3) to determine their impact on soil moisture retrievals. We investigated the weather-related surface condition effects of frozen conditions, snow, intercepted rain and dew, which are represented by nearby meteorological measurements, on Sentinel-1 $\sigma^0$ observations collected over five meadows, five cultivated fields and five forests in the Twente region in the east of the Netherlands. The results from these analyses are used to define a set of rules that could be taken as a starting point for the development of a formal masking procedure. After the application of the developed masking rules, the Sentinel-1 $\sigma^0$ observations of the five forests are used to estimate Sentinel-1's radiometric uncertainty ($s_{S1}$) as a function of the forest surface area ($A$) over which the Sentinel-1 $\sigma^0$ observations are averaged (from 0.25 ha to 10 ha). With the quantifications of $s_{S1}$ and the disturbing effects on $\sigma^0$ by the weather-related surface conditions, we determined their impact on soil moisture retrievals from Sentinel-1 $\sigma^0$ for various bare surface conditions. The widely applied Integral Equation Method (IEM) surface scattering model [36] was employed to simulate the $\sigma^0$ sensitivity to soil moisture for a surface representing meadows and a surface representing cultivated fields, for the incidence angles at which Sentinel-1 observes the Twente region and for dry to wet soil conditions. With these analyses we aim to provide insight in the role that the $s_{S1}$ and the weather-related surface conditions (if not masked) play in the accuracy of soil moisture retrievals from Sentinel-1 $\sigma^0$.

## 2. Study Area and Data

### 2.1. Twente Region and Meteorological Measurements

The Twente region, located in the eastern part of the Netherlands (Figure 1a), has a temperature oceanic climate. In the region are 20 stations equipped with 5TM soil moisture and soil temperature sensors [37] at depths of 5 cm, 10 cm, 20 cm, 40 cm and 80 cm [38,39].

The Twente region is rather flat with some elevated glacial ridges, see Figure 1b. The land cover in the region is a mosaic of cultivated crop fields, meadows, forested and built-up areas. The masking rules for weather-related surface conditions have been developed on Sentinel-1 $\sigma^0$ observations over five forests, five meadows and five cultivated fields (Figure 1b), and the $s_{S1}$ has been estimated using the $\sigma^0$ observations of the five forests. These forest areas (Figure 1c–g) were selected because they consist of dense forest, they are approximately homogeneous and have an area larger than 10 ha. The CORINE Land Cover maps 2012 and 2018 [40] and the High Resolution Layer Forest 2012 and 2015 [41] show that forest I–III are coniferous forest. Forest IV is dominated by coniferous forest (classified as coniferous forest in the CORINE Land Cover maps) and forest V contains a significant portion of deciduous forest (classified as deciduous forest in the CORINE Land Cover maps). Field visits have confirmed these classifications and revealed that most parts of the forests I–V have an understory of deciduous bush and trees, and that the soil surfaces are covered by litter.

According to laboratory analyses [39] and the soil physical map of the Netherlands (BOFEK2012 [42]), sandy to loamy sandy soils dominate the surface layer in the Twente region. The average soil texture from BOFEK2012 [42] for the selected meadows and cultivated fields is 81.5 % sand, 14 % silt and 4.5 % clay, with a bulk density of 1.36 g cm$^{-3}$.

The Royal Netherlands Meteorological Institute ("Koninklijk Nederlands Meteorologisch Instituut", KNMI) operates three automated weather stations in the Twente region (Figure 1b). The stations provide hourly measurements [43], of which air temperature (1.5 m above ground) and relative humidity (1.5 m above ground) at hourly time steps, rainfall as hourly sums and wind speed (10 m above ground) as hourly averages were used to develop the masking rules for frozen conditions, rain interception and dew (see Section 3.2). Table 1 lists some statistics on the measurements by the KNMI weather stations in the Twente region.

Adjacent to one of the selected meadows (since 13 April 2017) and two of the selected cultivated fields (since 27 May 2016 and 13 April 2017) rainfall was measured by a tipping bucket rain gauge (Davis Rain Collector 7857M [44]) with a resolution of 0.2 mm, also shown in Figure 1b. If available, these measurements were used to develop the masking rule for rain interception (see Section 3.2.3).

KNMI also operates a network of precipitation stations (Figure 1b) that record precipitation and snow depth daily at 09:00 CET (Central European Time) [43]. These measurements were used to develop the masking rule for snow (see Section 3.2.2). Over 2014–2019 there were on average 10 days per year that the surface was covered by snow, varying between 5 days in the hydrological year 2015 (1 April 2015–31 March 2016) and 23 days in the hydrological year 2016 (1 April 2016–31 March 2017).

**Table 1.** Summary statistics of the measurements by the three KNMI weather stations [43] in the Twente region, averaged over five hydrological years (1 April 2014–31 March 2019).

| Condition | Twenthe | Hupsel | Heino |
|---|---|---|---|
| Minimum monthly mean air temperature [°C] | 2.9 (January) | 3.2 (January) | 3.0 (January) |
| Maximum monthly mean air temperature [°C] | 19.0 (July) | 19.0 (July) | 18.7 (July) |
| Number of days per year on which the air temperature dropped below 0 °C | 53 | 49 | 49 |
| Annual amount of rainfall [mm] | 830 | 744 | 740 |
| Number of days per year with rain (minimum 1 mm d$^{-1}$) | 139 | 124 | 131 |

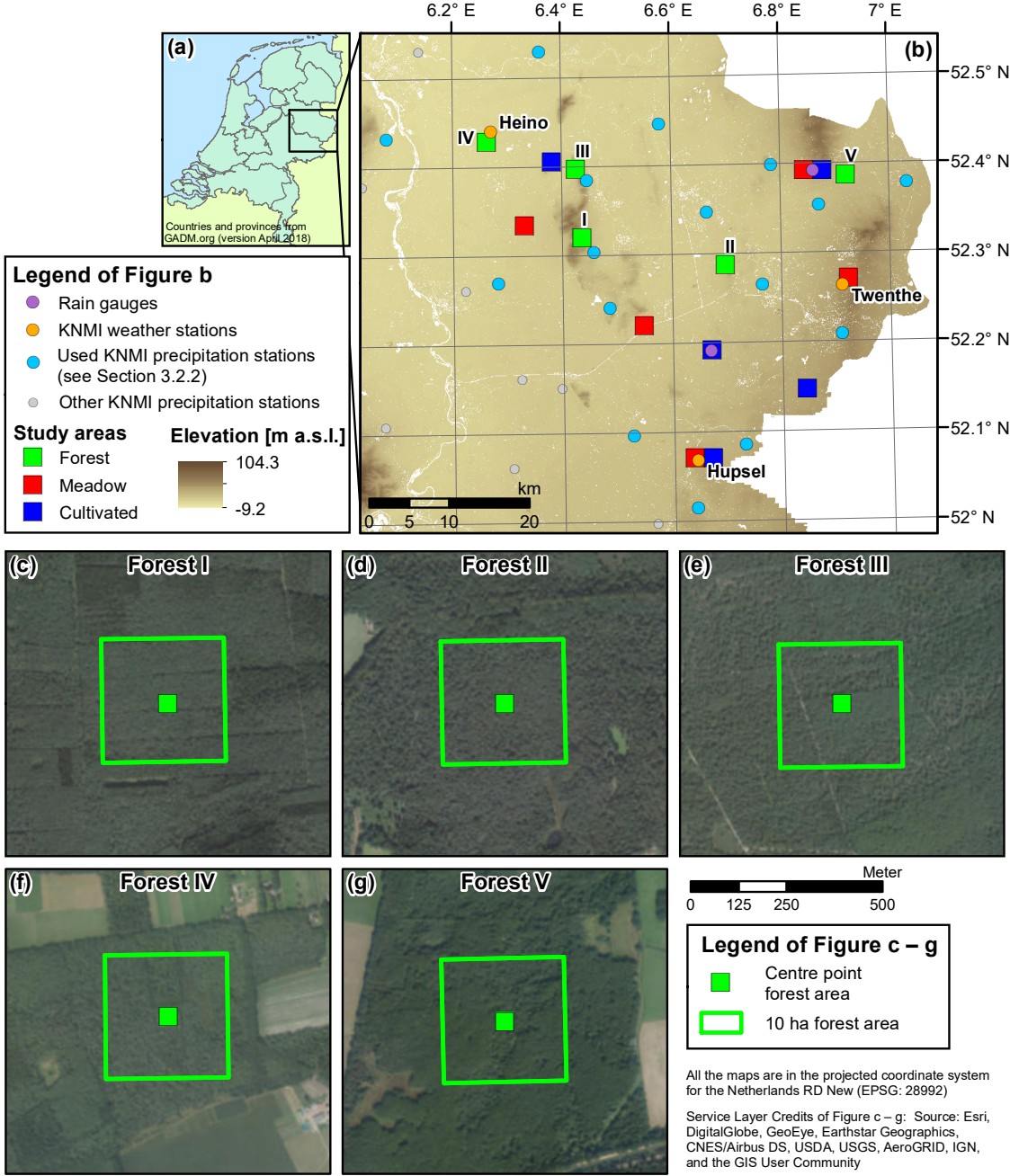

**Figure 1.** (**a**) Location of the Twente region in the Netherlands. (**b**) The Twente region with the selected study fields and the meteorological measurement locations. Background is the digital elevation model AHN2 [45]. The grid shows WGS84 coordinates. (**c–g**) Zoom-ins of the selected forest areas I to V.

## 2.2. Sentinel-1 Imagery

The Sentinel-1 constellation (Sentinel-1A and Sentinel-1B) provides over land images in Interferometric Wide Swath (IW) mode at VV and VH polarization, C-band (5.405 GHz), with a reported radiometric accuracy of 1 dB (three standard deviations) and, after multi-looking, a spatial resolution of 20 m × 22 m (4.4 equivalent number of looks) for the Ground Range Detected (GRD) High Resolution (HR) product [7,46]. By analysing timeseries of Sentinel-1 $\sigma^0$ observations from point targets with a well-specified $\sigma^0$, radiometric accuracy standard deviations of 0.30 dB (one standard deviation) for Sentinel-1A [9], and 0.29 dB (one standard deviation) for Sentinel-1B [8] have been obtained.

We processed Level-1 GRD HR IW Sentinel-1 images [47] using the following operations available in ESA's Sentinel Application Platform (SNAP) V6.0 [48]: (1) Apply Orbit File, (2) Thermal Noise Removal, and (3) Range Doppler Terrain Correction, including radiometric normalization to $\sigma^0$ (in $m^2\,m^{-2}$) with projected local incidence angles (see Table 2) on a geographic grid (WGS84) with a pixel spacing of 9E-5° (equivalent to $10\,m \times 6.1\,m$ at the study region's latitude). Subsequently, we averaged the Sentinel-1 $\sigma^0$ observations separately over the five individual forests for surface areas ranging from 0.25 ha (41 pixels) to 10 ha (1626 to 1631 pixels) taken from the centre points (see Figure 1c–g), the five individual meadows and the five individual cultivated fields, and we converted the intensity values to decibel (dB). Sentinel-1A (since 3 October 2014) and Sentinel-1B (since 28 September 2016) together make observations of the Twente region with a temporal resolution of 1.5 days. Table 2 specifies the orbits that cover the study region. We included all Sentinel-1 images available for the Twente region from 3 October 2014 until 1 November 2018, which are in total 676 images.

**Table 2.** Specifications of the Sentinel-1 orbits that cover the Twente region.

| Relative Orbit | Pass Direction | Acquisition Time (CET) | Number of Sentinel-1 Images | Projected Incidence Angle over the Study Fields |
|---|---|---|---|---|
| 15 | Ascending | 18:16 | 165 | 32.8°–36.8° |
| 37 | Descending | 6:49 | 166 | 33.7°–38.4° |
| 88 | Ascending | 18:24 | 171 | 41.2°–45.1° |
| 139 | Descending | 6:41 | 174 | 41.5°–46.2° |

## 3. Methods

### 3.1. Sentinel-1 Seasonal Anomalies

Seasonal $\sigma^0$ dynamics, if not removed from the $\sigma^0$ timeseries, would obscure the assessment of the effects of weather-related surface conditions on Sentinel-1 $\sigma^0$ and would cause an overestimation of the $s_{S1}$.

The $\sigma^0$ observations from meadows and cultivated fields vary due to variations in soil moisture and vegetation. As the $\sigma^0$ increases with increasing soil moisture (e.g., [5,6,49]), the Sentinel-1 $\sigma^0$ observations over the meadows and cultivated fields are expected to be higher in winter than in summer. The soil moisture measurements at $5\,cm$ depth in the Twente region indicate that the soil moisture is generally at a high level from mid-November to mid-March (mean is $0.42\,m^3\,m^{-3}$) and at a low level from mid-May to mid-October (mean is $0.23\,m^3\,m^{-3}$). However, the exact surface soil moisture cycle is different each year and also contains dynamics on shorter timescales in response to meteorological events. The growing season typically starts in April and ends in October, during which the meadows are cut multiple times and/or being grazed and the crops on the cultivated fields are sowed and harvested. Depending on the crop type and the development stage, the vegetation decreases the $\sigma^0$ when attenuation is dominant or increases the $\sigma^0$ when the scattering from the vegetation and the soil-vegetation pathways are dominant (e.g., [5,6,50]).

Although the Sentinel-1 signal is unlikely to penetrate forest canopies, forest $\sigma^0$ might still vary seasonally [12,25,26] because canopy development in the leafy period leads to an increase in foliar and stem biomass, and to changes in the vegetation's dielectric properties [26]. On top of the seasonal development in forest biomass, the leave and trunk water content (and thus the vegetation's dielectric properties) may vary in response to the hydrological conditions [51,52]. For example, Frolking et al. [52] identified a decrease in Ku-band $\sigma^0$ observations over the southwestern Amazon forest during a prolonged drought.

In addition to the $\sigma^0$ variations in response to changes in land surface parameters, there may be instabilities in the Sentinel-1 $\sigma^0$ observations over time [9,10]. For example, El Hajj et al. [10] reported an offset of +0.9 dB in Sentinel-1A $\sigma^0$ observations over three time-invariant targets (an asphalt racetrack,

a pine tree forest and a tropical forest) between 19 March 2015 and 25 November 2015 compared to observations outside this period, which they attributed to changes in the calibration of the Sentinel-1 products. In the period from 25 November 2015 to August 2017 the Sentinel-1A $\sigma^0$ observations varied between −0.75 dB (in October 2016) and +0.6 dB (in May 2016) relative to the $\sigma^0$ that is expected from point targets [9], which Schmidt et al. [9] explained by seasonal changes of the atmospheric attenuation and temporal trends of transmission and receive modules in the instrument's front-end.

To remove the seasonal variation in Sentinel-1 $\sigma^0$ observations, we considered the anomalies of Sentinel-1 $\sigma^0$ observations with the moving average that resembles their seasonality ($d\sigma^0$, in dB):

$$d\sigma^0(t) = \sigma^0(t) - \frac{1}{N} \sum_{i=t-W}^{i=t+W} \sigma^0(i), \tag{1}$$

where $t$ is an observation number in the $\sigma^0$ timeseries, $W$ is the length of the moving average window applied both forwards and backwards and $N$ is the number of observations in the window from $t - W$ to $t + W$.

At the moving average window that resembles the seasonality of the $\sigma^0$ timeseries, anomalies are assumed to be exclusively due to measurement noise and speckle (i.e., random). Therefore, at this window length the autocorrelation of the $d\sigma^0$ timeseries should be 0. A too small window will result in a negative autocorrelation of $d\sigma^0$, because the moving average will tend to follow the $\sigma^0$ timeseries and a negative anomaly will likely be followed by a positive anomaly (and vice versa). As a consequence, part of the uncertainty in which we are interested is removed. A too large window will result in a positive autocorrelation of $d\sigma^0$, because the moving average underestimates the seasonal dynamics of $\sigma^0$. To find the moving average window that resembles the seasonal dynamics of $\sigma^0$, we evaluated the autocorrelation at a lag of 1 for varied window lengths (the results are in Section 4.1).

### 3.2. Sentinel-1 Masks for Weather-Related Surface Conditions

Masks for frozen conditions, snow, rain interception and dew were developed by comparing meteorological measurements that represent these surface conditions against the Sentinel-1 $d\sigma^0$ of the five forests (the 10 ha forest areas), the five meadows and the five cultivated fields. We used standard meteorological measurements, which adds to the general applicability of the resulting masking rules. The results are presented in Section 4.2.

#### 3.2.1. Frozen Conditions

Both the water in the soil and in the vegetation can freeze, which we together refer to as frozen conditions. Its effect on Sentinel-1 $\sigma^0$ observations was investigated by comparing Sentinel-1 $d\sigma^0$ against the linear interpolation result of the two hourly air temperature measurements that are closest in time to a Sentinel-1 overpass, made by the KNMI weather station nearest to a study field.

#### 3.2.2. Snow

For the assessment of the effect of snow on Sentinel-1 $d\sigma^0$, we averaged the snow depth measurements of the three KNMI precipitation stations nearest to a study field. Subsequently, the daily snow depth measurements at 9:00 CET were linearly interpolated to the Sentinel-1 overpass times (listed in Table 2).

#### 3.2.3. Rain Interception

Local rain gauge measurements are available for one meadow and two cultivated fields. For rainfall at the other study fields we used the hourly measurements from the nearest KNMI weather station. To denote the possibility of rain interception on the surface and the canopy, we summed the precipitation in the hour in which the Sentinel-1 observation was acquired and the 12 preceding

hours following Cisneros Vaca and Van der Tol [25] for a coniferous and a deciduous forest in the Netherlands. Iida et al. [53,54] also used a period of 12 h to separate rainfall interception events.

### 3.2.4. Dew

Dew consists of water droplets that have condensed from the air on cooler objects, such as leaves [27,55]. Dew is not measured directly in the Twente region, but it is known that relative humidity conditions larger than 90 % are optimal for dew formation [27].

Rao et al. [56] showed that a relative humidity threshold of 90 % is a rather good predictor of dew onset and duration on maize, and they found only limited improvement with physically-based models. Because of the limited availability of the variables that are required as input to physically-based models [56], we opted for using relative humidity as an estimation for the likelihood of dew. The two hourly relative humidity measurements closest in time, made by the KNMI weather station nearest to a study field, were linearly interpolated to a Sentinel-1 overpass time.

Dew most likely forms during the night/morning and other stimuli are low wind speeds of about $1 \, \mathrm{m \, s^{-1}}$ to $2 \, \mathrm{m \, s^{-1}}$, clear skies and a temperature gradient between the object and the ambient atmosphere of about $1 \, ^\circ \mathrm{C}$ to $2 \, ^\circ \mathrm{C}$ [27,55,57]. Gleason et al. [57] used a threshold of $2.5 \, \mathrm{m \, s^{-1}}$ for wind speed (at $10 \, \mathrm{m}$ above the ground) below which dew is more likely to form. Based on these studies, we analysed the effect of relative humidity on $d\sigma^0$ for the morning and the evening overpasses of Sentinel-1 and for wind speeds below and above $2.5 \, \mathrm{m \, s^{-1}}$.

### 3.3. Sentinel-1 Radiometric Uncertainty

After masking the Sentinel-1 $\sigma^0$ timeseries, we assume that the remaining anomalies in the forest $d\sigma^0$ timeseries are due to Sentinel-1 measurement noise and speckle. The standard deviation of the $d\sigma^0$ timeseries is a measure for the radiometric uncertainty ($s_{S1}$, in dB), formulated as:

$$s_{S1} = \sqrt{\frac{\sum_{i=1}^{i=N}(d\sigma^0(i) - \overline{d\sigma^0})^2}{N-1}}, \qquad (2)$$

where $\overline{d\sigma^0}$ is the mean of the $d\sigma^0$ timeseries (in dB) and $N$ is the total number of observations in the timeseries.

We estimated $s_{S1}$ with Equation (2) for the five forests for surface areas ($A$) ranging from $0.25 \, \mathrm{ha}$ to $10 \, \mathrm{ha}$. Subsequently, a second-order power function between $A$ (in ha) and $s_{S1}$ (in dB) of the following form was fitted:

$$s_{S1} = c_1 A^{c_2} + c_3. \qquad (3)$$

The first term on the right-hand side of Equation (3) describes the dependency of $s_{S1}$ to $A$. According to the Rayleigh fading model for speckle, each $\sigma^0$ observation is a sample from a Rayleigh distribution [6]. By averaging independent $\sigma^0$ observations, like spatially averaging them over an area ($A$), the uncertainty of the mean $\sigma^0$ decreases with $1/\sqrt{n}$, where $n$ is the number of independent samples [6]. This follows from the standard deviation of a sample mean which decreases with $1/\sqrt{n}$, as is also formulated as part of the central limit theorem [58]. Thus, the expected value for $c_2$ is $-0.5$ ($A^{-0.5} = 1/\sqrt{A}$). For a very large $A$ (infinite number of samples) the uncertainty due to speckle approaches $0 \, \mathrm{dB}$. The second term on the right-hand side of Equation (3) specifies the $s_{S1}$ which is present due to the variation in a $\sigma^0$ timeseries as a result of inherent sensor instabilities and calibration uncertainties. The model coefficients $c_1$ to $c_3$ were obtained by the 'Trust-Region' algorithm in the Matlab Curve Fitting toolbox. The results are presented in Section 4.3.

### 3.4. Impact on Soil Moisture Retrieval Accuracy

The physically-based IEM surface scattering model [36], of which various versions have widely been used (e.g., [1–3,5,20,49,59–64]), simulates the $\sigma^0$ of bare land surfaces as a function of the land

surface's geometric and dielectric properties. For more background on the IEM model, readers are referred to Ulaby et al. [6]. We employed IEM to simulate the $\sigma^0$ sensitivity to soil moisture for various bare surface conditions.

The geometry of the land, also known as the surface roughness, is parameterized by the root mean square surface height ($z$), the autocorrelation length ($c_l$) and an autocorrelation function. An exponential autocorrelation function was selected here, because it is viewed as most applicable to agricultural fields [6,59]. Table 3 lists the surface roughness scenarios that were investigated, representing the surfaces of meadows and cultivated fields. The found surface roughness parameters describe a slightly smoother surface for meadows than for cultivated fields. A very smooth surface scenario was added for reference purposes.

The land surface's dielectric properties, composed of the real and imaginary relative permittivity ($\epsilon_r$), were estimated with the Mironov soil dielectric mixing model [65] using the average soil texture of the meadows and cultivated fields (see Section 2.1) and soil moisture as input. For the impact of the weather-related surface conditions on soil moisture retrievals, we evaluated a soil moisture range of $0.05\,\mathrm{m^3\,m^{-3}}$ to $0.50\,\mathrm{m^3\,m^{-3}}$. For the impact of the $s_{S1}$ on soil moisture retrieval uncertainty, we evaluated a dry scenario (soil moisture equal to $0.10\,\mathrm{m^3\,m^{-3}}$) and a wet scenario (soil moisture equal to $0.35\,\mathrm{m^3\,m^{-3}}$).

The $\sigma^0$ sensitivity to soil moisture was simulated with IEM for the three surface roughness scenarios (Table 3) and the soil moisture conditions as defined above, for incidence angles of 35° (representative for Sentinel-1 orbits 15 and 37, see Table 2) and 44° (representative for Sentinel-1 orbits 88 and 139, see Table 2). Figure 2 illustrates the methodology that was used to calculate the impact of a deviation in $\sigma^0$ ($\Delta\sigma^0$) on soil moisture retrievals: $\Delta\sigma^0$, either due to $s_{S1}$ (both plus and minus) or a disturbing surface condition (single direction), is superimposed on the $\sigma^0$ for a specific soil moisture value ($\theta$). The difference between the soil moisture that is retrieved with $\sigma^0 \pm \Delta\sigma^0$ and the soil moisture starting point ($\theta$) resembles the impact on the soil moisture retrieval ($\Delta\theta^\pm$). The performance of soil moisture retrievals is typically defined as the standard deviation of the differences between retrievals and a reference, i.e., the unbiased $E_{RMS}$ [32]. The standard deviation is also taken as a measure for the $s_{S1}$. Hence, the $\Delta\theta^\pm$ that is obtained from $s_{S1}$ can be considered equivalent to the unbiased $E_{RMS}$ (both are in $\mathrm{m^3\,m^{-3}}$).

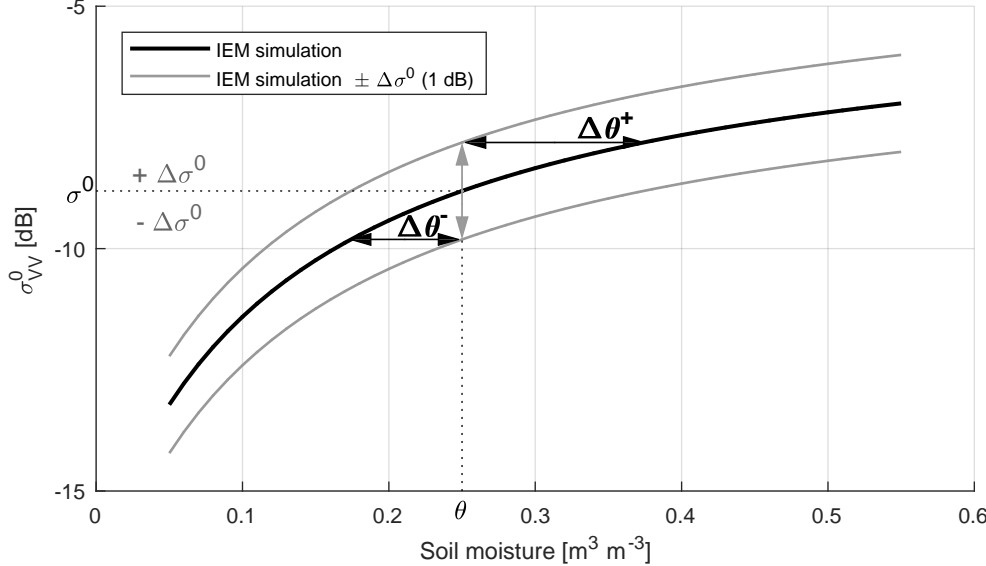

**Figure 2.** Illustration of the method to estimate the impact on soil moisture retrievals ($\Delta\theta$) at a specific soil moisture value ($\theta$) due to a deviation ($\Delta\sigma^0$) in VV backscatter ($\sigma^0_{VV}$), for a certain relation between $\sigma^0_{VV}$ and soil moisture simulated by IEM. This example uses the surface roughness parameters representative for meadows, listed in Table 3.

**Table 3.** The root mean square surface height ($z$) and the autocorrelation length ($c_l$) for three surface roughness scenarios. These surface roughness scenarios were used to evaluate the impacts of the weather-related surface conditions and $s_{S1}$ on soil moisture retrievals.

| Scenario | $z$ [cm] | $c_l$ [cm] | Source |
|----------|----------|------------|--------|
| Cultivated fields | 1.1 | 11.5 | Average of the measurements for corn fields during SMAPVEX-12 [66] |
| Meadows | 0.94 | 14.8 | Average of the measurements for pastures during SMAPVEX-12 [66] |
| Very smooth surface | 0.5 | 10 | Added for reference purposes |

IEM can simulate the $\sigma^0$ for the VV polarization and the VH polarization. In Section 4.4 we show the results for the VV polarization, because this channel is generally used in soil moisture retrieval procedures for its larger sensitivity to soil moisture (e.g., [1,4]).

## 4. Results and Discussion

### 4.1. Sentinel-1 Timeseries and Seasonal Anomalies

Figure 3 shows the effect of varied moving average windows in Equation (1) on the standard deviation and autocorrelation of the $d\sigma^0$ timeseries to find the moving average window that resembles the seasonal dynamics in the Sentinel-1 $\sigma^0$ timeseries. The standard deviation is larger when no moving average is applied (Figure 3a,b), so the $s_{S1}$ would be overestimated when the $\sigma^0$ seasonal dynamics are not removed. This is explained by the variations in the timeseries due to seasonal dynamics and instabilities in the Sentinel-1 $\sigma^0$ observations that are still included, which is also reflected in the autocorrelation being significantly above 0.0 (Figure 3c,d). Regarding forests the autocorrelation is approximately 0.0 for a moving average window of 40 days. A moving average window of 40 days backwards and 40 days forwards is close to the three-month timescale of the meteorological seasons in the Twente region. Regarding meadows and cultivated fields the autocorrelation is approximately 0.0 for a moving average window of 25 days. The smaller window for meadows and cultivated fields can be explained by the shorter growth cycles of vegetation and harvesting and the shorter timescales of soil moisture dynamics, which strongly affect the $\sigma^0$ from these fields.

Figure 4 shows examples of Sentinel-1 $\sigma^0$ and $d\sigma^0$ timeseries for a forest, a meadow and a cultivated field. For brevity only the VV polarization is shown. The Sentinel-1 $\sigma^0$, moving average and $d\sigma^0$ timeseries of all the study fields for both the VV and VH polarization are included in Supplement 1.

The variation in the moving average (the second term on the right-hand side of Equation (1)) timeseries of the forest $\sigma^0$ is limited compared to the results from the meadows and cultivated fields. Excluding the Sentinel-1 observations that are masked for frozen conditions, snow or rain interception in the calculation of the moving averages, the minimum-to-maximum ranges of the moving average for the five forests are 1.04 dB–2.09 dB (mean is 1.57 dB) for the VV polarization and 0.91 dB–2.68 dB (mean is 1.52 dB) for the VH polarization. The mean minimum-to-maximum moving average range is 5.35 dB for the meadows and 8.57 dB for the cultivated fields for the VV polarization, and 4.83 dB and 9.17 dB, respectively, for the VH polarization. For C-band SAR observations, Cisneros Vaca and Van der Tol [25] and Dobson et al. [26] showed that summer conditions lead to an increase of 0.7 dB–1 dB in the $\sigma^0$ from a coniferous Douglas-fir forest and a coniferous pine forest, respectively. For a deciduous beech forest Cisneros Vaca and Van der Tol [25] showed an increase of 0.5 dB at the VV polarization and a decrease of 1 dB at the VH polarization, whereas Dobson et al. [26] identified a general decrease of 0 dB–2 dB for a deciduous forest. The seasonal variations found in these studies are of the same order of magnitude as the moving average ranges that were estimated for the five selected forests. In Section 3.1 we also discussed an offset of +0.9 dB in Sentinel-1A $\sigma^0$ observations between 19 March 2015 and 25 November 2015 [10]. Indeed, forest $\sigma^0$ increases in this period (see Figure 4a and Supplement 1), although part of the $\sigma^0$ offset will be due to the seasonal effect on $\sigma^0$. In the further analyses in this

study these seasonal dynamics are removed from the $\sigma^0$ timeseries by considering the anomalies with the moving averages ($d\sigma^0$).

As can be seen in the examples in Figure 4 and deduced from Figure 3a,b, $d\sigma^0$ values are generally smaller for the forests than for the meadows and cultivated fields. The larger $d\sigma^0$ for meadows and cultivated fields relate to the effects of short-term variations in soil moisture and abrupt changes in vegetation, which superimpose on the seasonality in Sentinel-1 $\sigma^0$ as determined by the moving average.

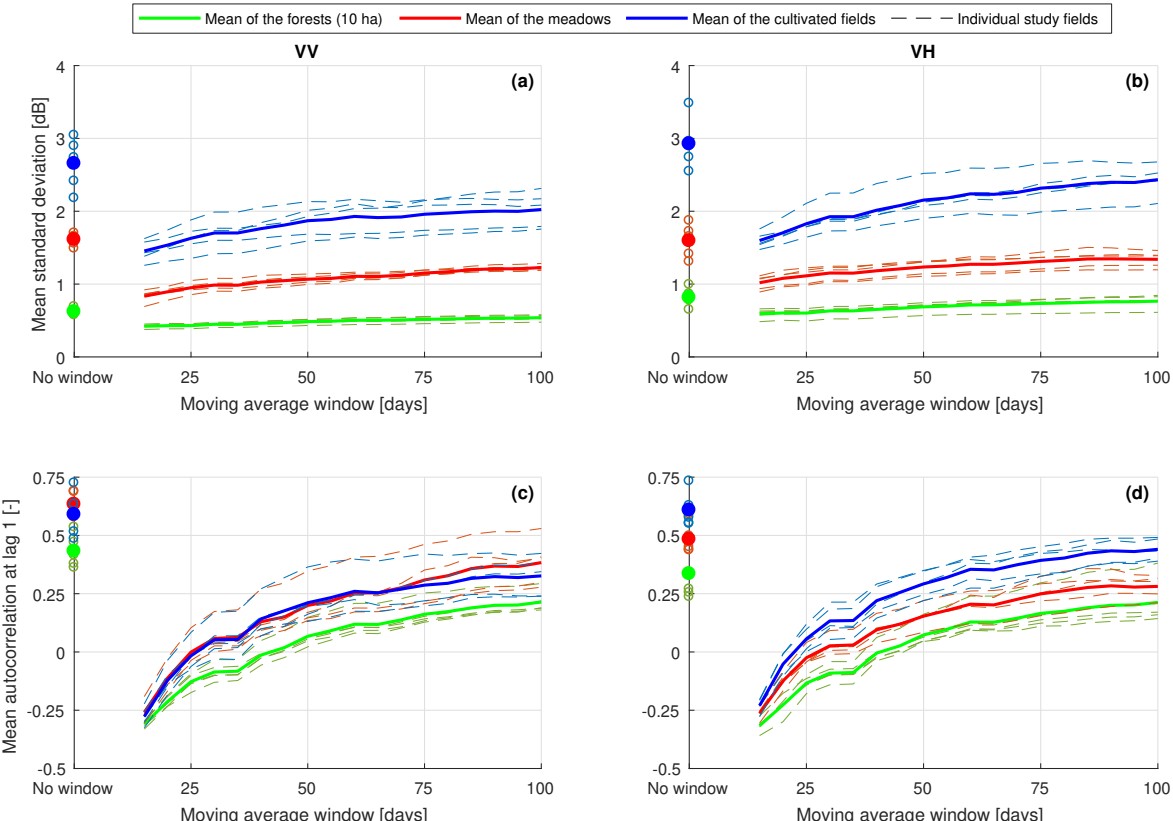

**Figure 3.** Standard deviation for the VV (**a**) and VH polarization (**b**) and autocorrelation at a lag of 1 for the VV (**c**) and VH polarization (**d**), averaged over all orbits, for the $d\sigma^0$ timeseries calculated with Equation (1) using a moving average window varying from 15 days to 100 days. The values labelled as 'no window' were calculated for the anomalies of Sentinel-1 $\sigma^0$ observations with respect to the mean of the timeseries. The dashed lines represent the individual study fields and the solid lines are the means per land cover type.

### 4.2. Effects by Weather-Related Surface Conditions

The effects of frozen conditions, snow, intercepted rain and dew on Sentinel-1 $d\sigma^0$ are discussed in Sections 4.2.1–4.2.4. The meteorological measurements that are used to represent the weather-related surface conditions are introduced in Section 3.2, and $d\sigma^0$ follows from Equation (1) and the analysis in Section 4.1.

### 4.2.1. Frozen Conditions

Figure 5 shows $d\sigma^0$ against air temperature. For the VV and the VH observations of the three land cover types, the bin means of 25 $d\sigma^0$ data points along the x-axis decrease with air temperature below approximately 1 °C. In other words, for low temperatures the Sentinel-1 observations are lower than their moving averages. This is also visible in Figure 4 in the majority of the Sentinel-1 observations that are masked for frozen conditions. From this analysis follows the masking rule for Sentinel-1

observations that were acquired when air temperature was below 1 °C, listed in Tables 4 and 5. The air temperature threshold of 1 °C could be explained by the measurement height of 1.5 m above ground, at which the temperature is generally higher than closer to the ground.

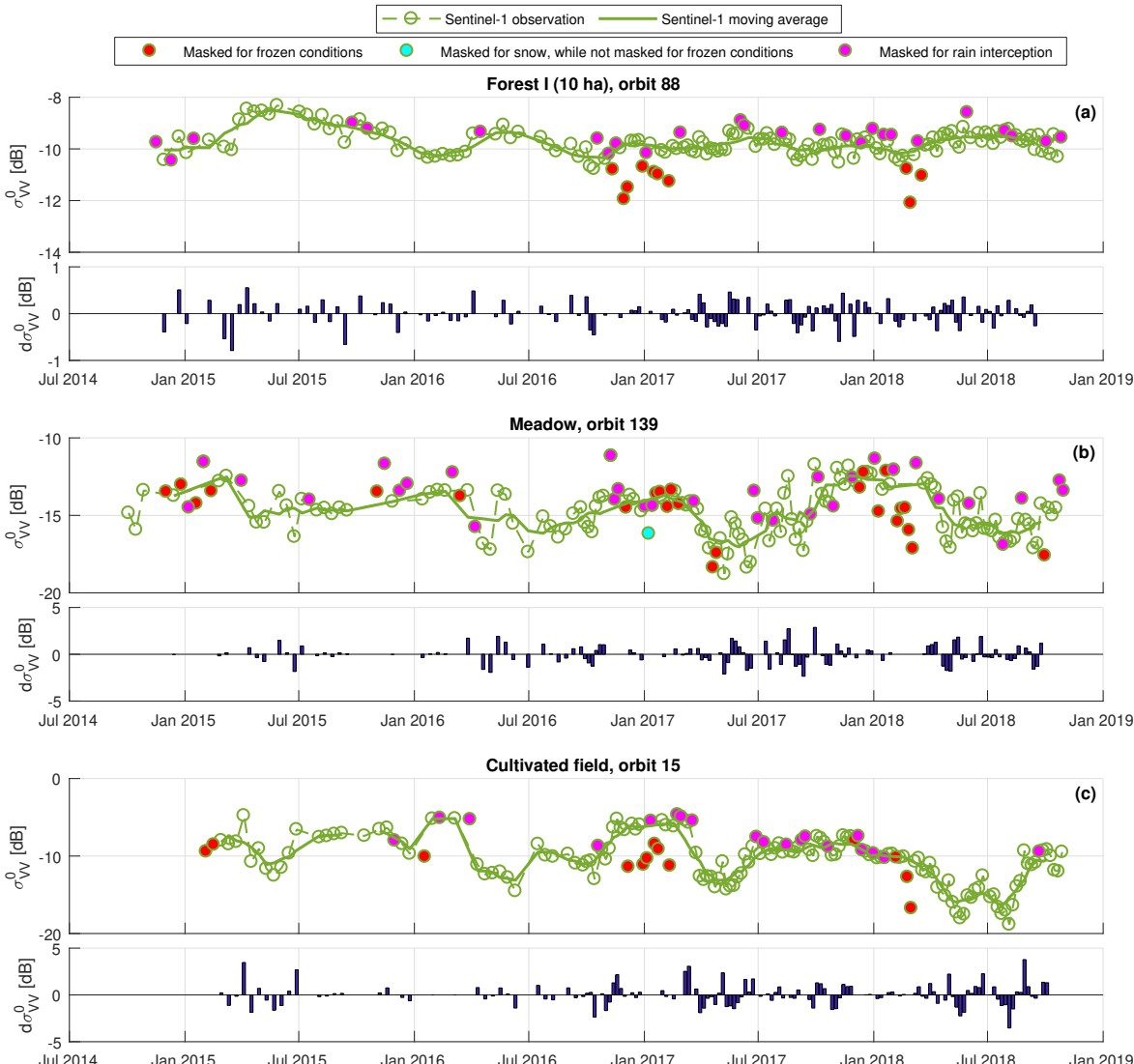

**Figure 4.** Sentinel-1 VV backscatter observations ($\sigma^0_{VV}$) and anomalies with the moving average ($d\sigma^0$) calculated with Equation (1) using a moving average window of 40 days for (**a**) an example forest and 25 days for (**b**) an example meadow and (**c**) an example cultivated field. The Sentinel-1 observations that are masked for frozen conditions, snow or rain interception are not included in the calculation of the moving averages. The masking rules are defined in Section 4.2.

Hallikainen et al. [21] and Mironov et al. [19] demonstrated that for wet soils the real and imaginary parts of $\epsilon_r$ are considerably lower in frozen than in thawed conditions, because the $\epsilon_r$ of ice is much lower than that of liquid water: $\epsilon_r \approx 3.2 - j0$ (ice) versus $\epsilon_r \approx 73.3 - j21.5$ (water at 20 °C) for C-band [6]. Consequently, the $\sigma^0$ is lower for frozen than for thawed land surfaces. Figure 5 and the mean effects on $d\sigma^0$ when the masking rule for frozen conditions applies (listed in Tables 4 and 5) indicate that the effect is stronger for the $\sigma^0$ observations in the VH than in the VV polarization, which was also found by Baghdadi et al. [20] for Sentinel-1 observations. The largest signal is observed for the cultivated fields.

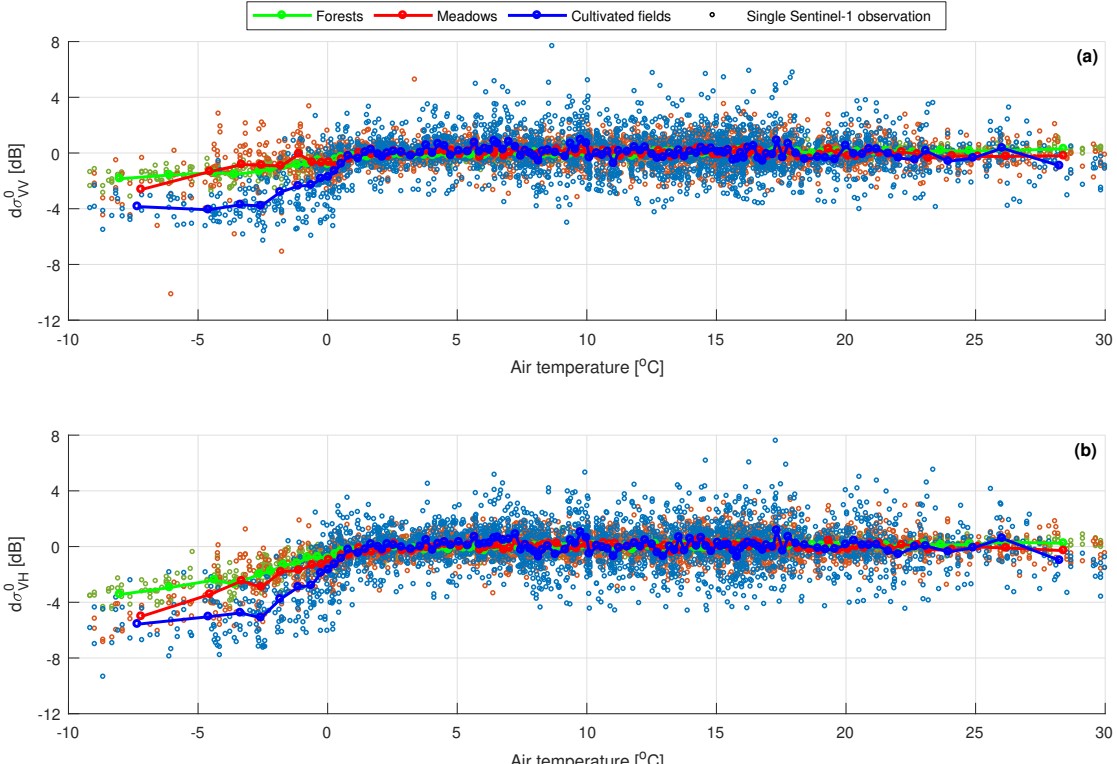

**Figure 5.** The effect of air temperature on the seasonal anomalies ($d\sigma^0$) of the Sentinel-1 $\sigma^0$ observations in VV polarization (**a**) and VH polarization (**b**). Each point on the lines represents a bin of 25 Sentinel-1 observations along the x-axis. The Sentinel-1 observations masked for snow or rain interception are not included.

### 4.2.2. Snow

Snow is generally shallow and short-lived (couple of days) in the Twente region, so the number of Sentinel-1 observations under conditions with snow and the observed snow depth are limited. Moreover, during many Sentinel-1 observations when snow was present the masks for frozen conditions or rain interception also apply. Therefore, in Figure 6 ($d\sigma^0$ against snow depth) and Figure 7 ($d\sigma^0$ against air temperature, when snow was present) we also show the Sentinel-1 observations that would actually be masked for rain interception and frozen conditions, respectively.

Neither the presence of snow, the depth of the snow layer nor preceding precipitation seem to have an unambiguous effect on the $d\sigma^0$ in Figure 6. Some Sentinel-1 $\sigma^0$ observations are clearly lower than their moving average (for example for a snow depth of 0.70 cm), whereas other observations seem unaffected (for example for a snow depth of 0.87 cm). Based on simulations with a radiative transfer model, Ulaby and Long [6] have deduced that a dry snow layer is optically thin for snowpacks below 2 m if $r/\gamma < 10^{-2}$, where $r$ is the ice-crystal radius and $\gamma$ is the wavelength. This condition holds for Sentinel-1's C-band, for typical ice-crystal radii of 0.25 mm to 2 mm [6]. In contrast, wet snow's $\epsilon_r$ is larger, and consequently wet snow has a larger reflectivity than dry snow [6]. Moreover, the absorption by wet snow is much larger than by dry snow, which also reduces the soil contribution to the $\sigma^0$ [6]. For a snowpack of 48 cm, HH $\sigma^0$ observations at a frequency of 5.5 GHz and an incidence angle of 50° reduce from −17 dB (snow liquid water content of 0 %) to −21 dB (snow liquid water content of 1.26 %) [67]. Using C-band HH (Radarsat) and VV (ERS SAR) observations and theoretical model simulations, Baghdadi et al. [23] and Nagler and Rott [22] also demonstrated that wet snow strongly reduces the $\sigma^0$ and both studies identified a change detection threshold of −3 dB compared to reference $\sigma^0$ from snow-free or dry snow surfaces for the classification of wet snow. As such, it is expected that dry snow does not affect the $\sigma^0$ and wet snow decreases the $\sigma^0$ observations.

Figure 7 plots $d\sigma^0$ against air temperature for the Sentinel-1 observations acquired in the morning and in the evening overpasses when snow was present. From Figure 7 it can be deduced that most of the large negative $d\sigma^0$ values in Figure 6 come from a Sentinel-1 morning overpass. The Sentinel-1 observations for the example with a snow depth of 0.70 cm (mentioned above in this section) were also acquired in a morning overpass, whereas the Sentinel-1 observations with a snow depth of 0.87 cm were acquired in an evening overpass.

When the average air temperature in the 6 h preceding a Sentinel-1 morning observation was above 0 °C and the snow depth at 9:00 CET shortly after the Sentinel-1 morning observation was 0 cm, it is likely that the snow at the time of the Sentinel-1 observation (at 6:45 CET) also already melted. Therefore, these Sentinel-1 morning observations are indicated with a different marker type in Figure 7a,c. Indeed, the $d\sigma^0$ values of those Sentinel-1 observations are around 0 dB. The remaining data points in Figure 7a,c, i.e. when a snow cover is assumed to be present, show that the meadow and cultivated field $\sigma^0$ observations acquired in the morning overpasses are disturbed even when the air temperature is above 1 °C. There are two possible explanations for this. Firstly, despite the air temperature being above 0 °C, the soil and the snow layer may still be frozen. This may be possible because the air temperature measurements are collected at 1.5 m above the ground surface. A frozen surface below the optically thin dry snow layer will reduce the $\sigma^0$. Secondly, the snow is melting (wet snow conditions). Both options would reduce the $\sigma^0$ compared to unfrozen and snow-free conditions, but it is impossible to distinguish them with the snow and temperature measurements available in this study. Baghdadi et al. [20] also noted that frozen soils and wet snow cannot reliably be distinguished with C-band $\sigma^0$ observations.

As with the effect of frozen conditions (see Section 4.2.1), the largest signal is observed for the cultivated fields. These fields are generally bare in winter. The Sentinel-1 forest $\sigma^0$ appears unaffected by snow when air temperature was above 1 °C. Baghdadi et al. [23] also found that, in contrast to i.a. alfalfa, forage crop, oat and grass, wet snow cannot be identified over forested areas, which they attributed to the attenuation of the ground $\sigma^0$ signal by the canopy. Some snow may reside on the forest canopy, but this will melt fast when the air temperature is above 0 °C.

The Sentinel-1 observations acquired in the evening overpasses (Figure 7b,d) are unaffected by snow when the air temperature was above 0 °C. The maximum observed snow depth is 1.3 cm, which is the time-interpolated result of the snow depth recorded at 9:00 CET before and after the Sentinel-1 observation at 18:20 CET. In reality it is likely that (most of) such shallow snow has already melted in the time between the snow depth measurement at 9:00 CET and the Sentinel-1 observation at 18:20 CET, when the average air temperature was above 0 °C. For most of the Sentinel-1 observations in Figure 7b,d the three closest KNMI precipitation stations also recorded no snow or only a broken snow cover at 9:00 CET the day after the Sentinel-1 observation. Even though the majority of the Sentinel-1 evening observations is evidently unaffected, some suspicious data points can be identified in Figure 7b,d. However, we could not find any evidence for the presence of snow, and therefore, considered these points as outliers.

From the analyses above follows that the Sentinel-1 morning observations of meadows and cultivated fields are affected by snow and should be masked. The masking rules for snow are summarized in Tables 4 and 5, along with the mean $d\sigma^0$ when the masking rule applies. For a further development of the snow mask, Sentinel-1 $\sigma^0$ observations should be analyzed in combination with detailed snow information, regarding e.g., depth, wetness and where the snow resides, for a region where snowpacks are deeper and long-lived.

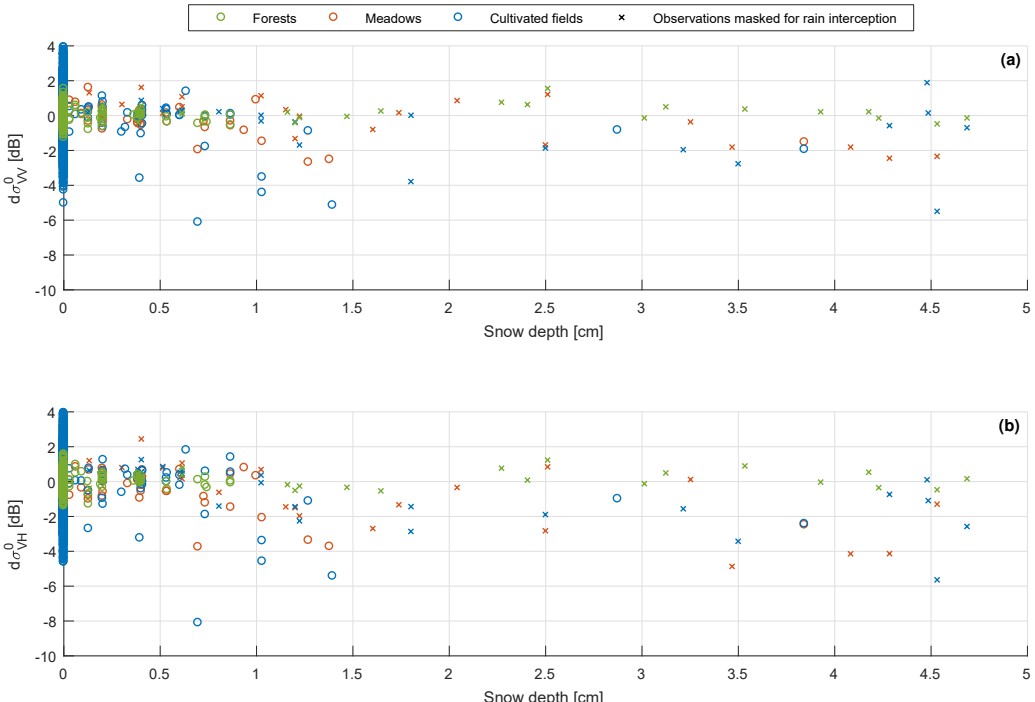

**Figure 6.** The effect of snow depth on the seasonal anomalies ($d\sigma^0$) of the Sentinel-1 $\sigma^0$ observations in VV polarization (**a**) and VH polarization (**b**). Each data point represents a single Sentinel-1 observation. The Sentinel-1 observations masked for frozen conditions are not included, whereas the Sentinel-1 observations that are masked for rain interception are included (marked with a cross).

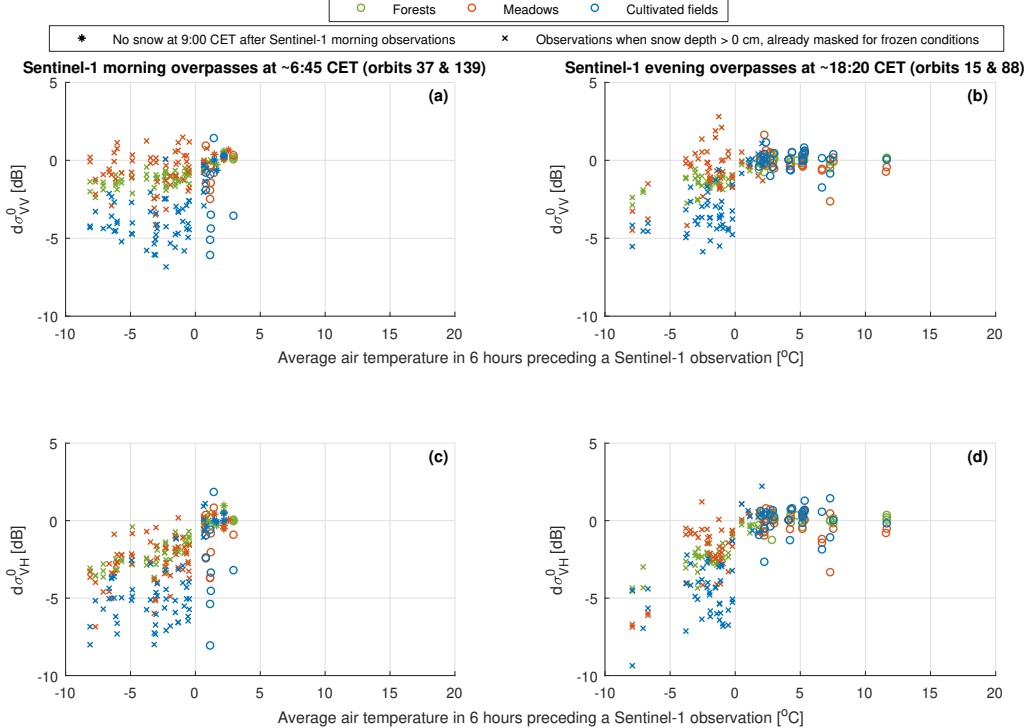

**Figure 7.** The effect of air temperature on the seasonal anomalies ($d\sigma^0$) when snow was present, for the Sentinel-1 VV $\sigma^0$ observations acquired in the morning overpasses (**a**) and the evening overpasses (**b**), and the Sentinel-1 VH $\sigma^0$ observations acquired in the morning overpasses (**c**) and the evening overpasses (**d**). Each data point represents a single Sentinel-1 observation. The Sentinel-1 observations masked for rain interception are not included, whereas the Sentinel-1 observations that are masked for frozen conditions are included (marked with a cross).

### 4.2.3. Rain Interception

Figure 8 shows that the $\sigma^0$ increases after rainfall in the 12 h preceding a Sentinel-1 observation. Regarding meadows and cultivated fields this will not exclusively being the effect of intercepted rain, but also the effect of increased soil moisture. Therefore, for the development of the rain interception mask we focused on the effect on $d\sigma^0$ that is observed for forests.

We defined the masking rule for rain interception as minimum 1.8 mm of rain (listed in Tables 4 and 5), because from this amount of rainfall the effect on the bin-averaged $d\sigma^0$ is consistently positive. Starting from 1.8 mm of rain, the $d\sigma^0$ values generally increase with the amount of rain until the bin-averaged $d\sigma^0$ values are approximately constant onwards from 4.2 mm of rain with an average effect of +0.47 dB for the VV polarization and +0.51 dB for the VH polarization. De Jong et al. [24] and Riedel et al. [29] explained that the $\sigma^0$ is influenced by the free-water content of the vegetation, and canopies intercept rainfall as a thin layer of free water on hydrophilic leaves or as drops on hydrophobic leaves [24]. Moreover, the orientation of the leaves can change due to the added weight [27]. Previous studies showed effects of rainfall on C-band $\sigma^0$ of +0.7 dB to +2.5 dB (mean is +1.3 dB) for a dense mixed forest [24], +1 dB to +2 dB for a Douglas-fir canopy [25], +2 dB to +3 dB for mature conifer and hardwood forests [26], and +1 dB for a pine tree forest [10]. Regarding crops, the X-band $\sigma^0$ observations by Allen and Ulaby (1984), as described by Jackson and Moy [27], show an increase of about 3 dB directly after the canopies (wheat, corn and soybeans) had been sprayed with water, whereas the C-band $\sigma^0$ observations by Riedel et al. [29] suggest that the effect on the $\sigma^0$ depends on the vegetation structure and the growth stadium.

Not only may the effect on the $\sigma^0$ by intercepted rain depend on the vegetation characteristics, also the amount and duration of the intercepted rain will vary. In this study we used the rainfall sum in the 12 h preceding a Sentinel-1 observation as a rather simple proxy for rain interception. In reality, many more factors, including evaporation, the canopy's density and shape of elements, and the timing and intensity of the event(s) preceding a Sentinel-1 observation, control whether a canopy is wet at the moment of a Sentinel-1 overpass. A rain interception model, such as the Gash model [68], should be used to simulate these effects, but this goes beyond the scope of this study.

Standing water on fields after a heavy rain event might also affect the Sentinel-1 $\sigma^0$ observations. Complete inundation of an area shows a clear decrease in the $\sigma^0$ (e.g., [17,18]), but standing water on an agricultural field is generally a mosaic of wet soil and standing water in local depressions. We have not collected information about standing water on the selected agricultural fields. Given the high infiltration capacity of the sandy to loamy sandy soils in the Twente region, we assume that the most severe standing water situations will already be masked out by the masking rule for rain interception based on a minimum of 1.8 mm of rain in the 12 h preceding a Sentinel-1 observation.

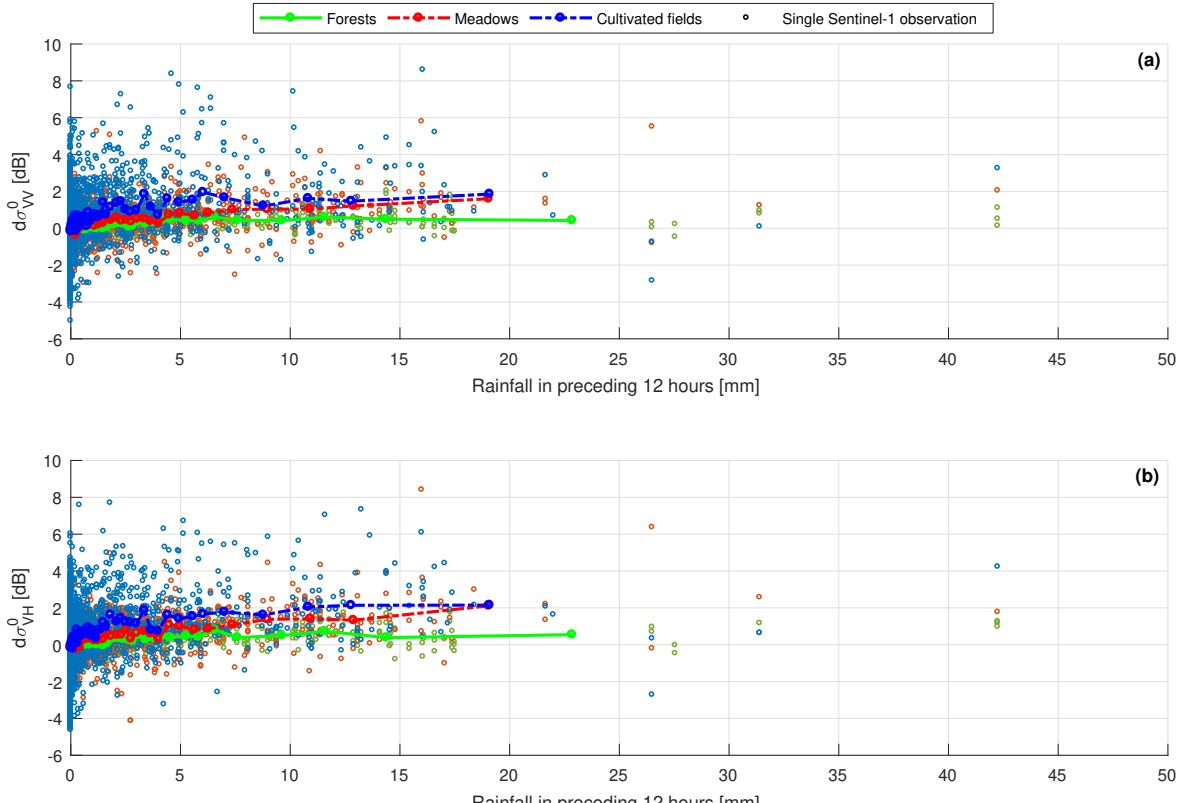

**Figure 8.** The effect of the rainfall sum in the preceding 12 h on the seasonal anomalies ($d\sigma^0$) of the Sentinel-1 $\sigma^0$ observations in VV polarization (**a**) and VH polarization (**b**). Each point on the lines represents a bin of 25 Sentinel-1 observations along the x-axis. The Sentinel-1 observations masked for frozen conditions or snow are not included.

### 4.2.4. Dew

Figure 9 shows $d\sigma^0$ against relative humidity for the Sentinel-1 morning and evening overpasses and for weak (wind speed $< 2.5\,\mathrm{m\,s^{-1}}$) and strong winds (wind speed $\geq 2.5\,\mathrm{m\,s^{-1}}$). In all figures $d\sigma^0$ seems unaffected by a high relative humidity. As explained in Section 3.2.4, the highest likelihood of dew is expected for the Sentinel-1 observations that were acquired during the morning overpasses when the wind speed was below $2.5\,\mathrm{m\,s^{-1}}$ and the relative humidity was above 90 %. Indeed, Figure 9 shows that the relative humidity is generally higher during Sentinel-1's morning overpasses, especially when the wind speed was low (Figure 9a,e). However, even Figure 9a,e exhibit no distinct systematic effect of high relative humidity ($> 90\,\%$) on $d\sigma^0$. In Figure 9a,e there is peak in the cultivated field bin-averaged $d\sigma^0$ of about 1 dB at a relative humidity of 98 %, but $d\sigma^0$ is not systematically positive from the expected threshold of 90 % and for relative humidity higher than 98 % the bin-averaged $d\sigma^0$ values decrease again. This is probably just inherent variation in the bin-averaged $d\sigma^0$ values, which is also observed for example at relative humidities of 65 % and 85 %. We could not find a systematic effect of dew on the Sentinel-1 $\sigma^0$ observations, so we do not mask them for dew (also see Tables 4 and 5).

This is supported by Jackson and Moy [27], who concluded, based on the work by Batlivala and Ulaby, that for frequencies between 1.1 GHz and 7.25 GHz there is no effect of dew on the $\sigma^0$. On the contrary, Gillespie et al. [28] did find an increase in $\sigma^0$ observations due to a dew event for C-band HH, VV and HV observations taken from incidence angles of 10° to 40° parallel to the crop row direction. However, the Ku- and L-band observations are less affected, and the C-band VH observations, at an incidence angle of 60° or in across-row direction seem unaffected by dew [28]. The C-band VV and VH $\sigma^0$ observations by Riedel et al. [29] decrease by about 0.5 dB to 2 dB due to a dew event, whereas the X-band observations slightly increase and the L-band observations respond differently for different

polarizations. These $\sigma^0$ observations show inconsistent $\sigma^0$ responses to dew for different radar settings and no explanations are provided for these differences [27], and Ulaby and Long [6] and Jackson and Moy [27] stated that no general conclusions can be drawn regarding the effects of dew as a function of frequency, angle and polarization and the mechanisms behind it.

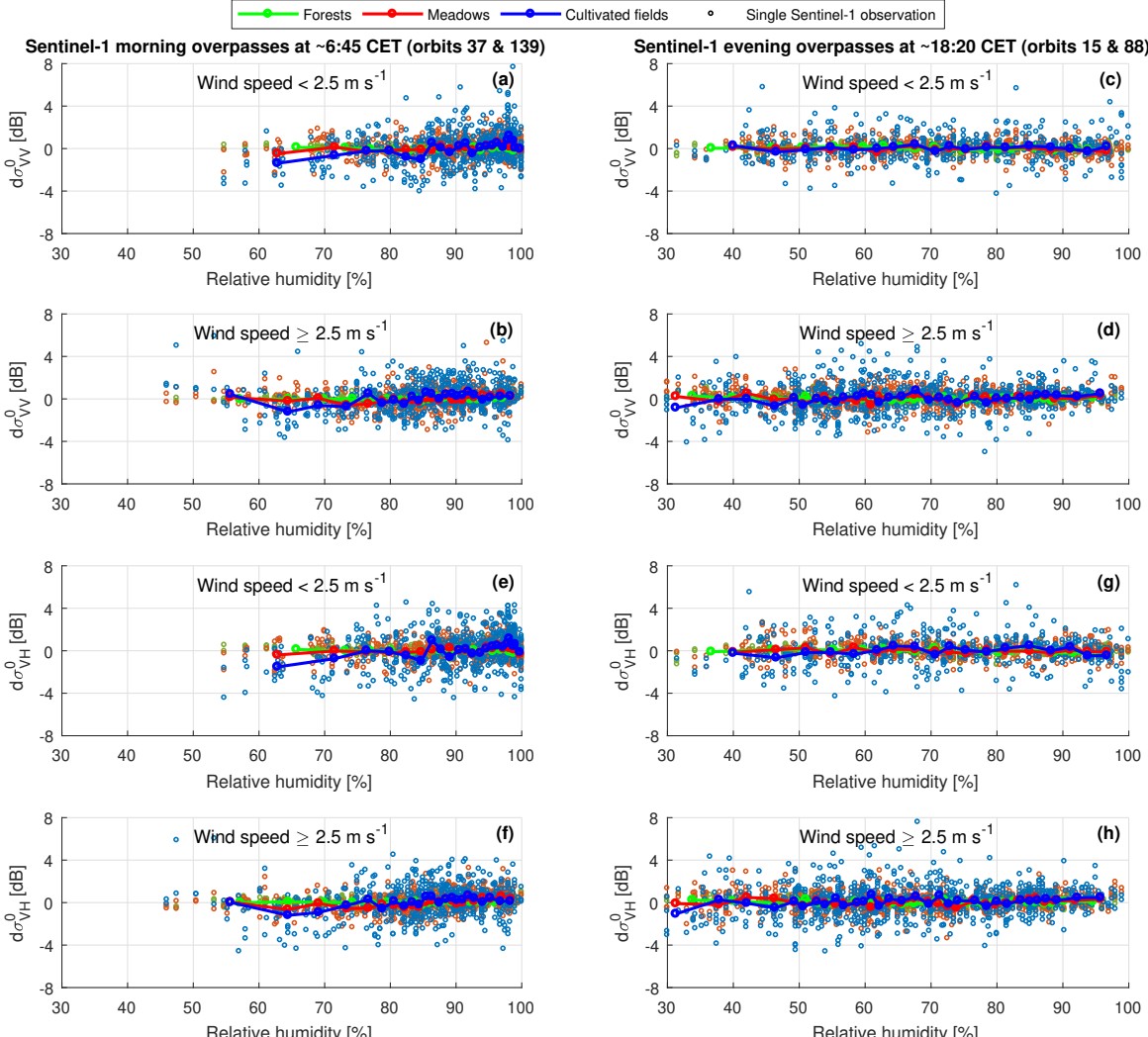

**Figure 9.** The effect of relative humidity on the seasonal anomalies ($d\sigma^0$), for the Sentinel-1 VV $\sigma^0$ observations acquired in the morning overpasses (**a**,**b**) and the evening overpasses (**c**,**d**), and the Sentinel-1 VH $\sigma^0$ observations acquired in the morning overpasses (**e**,**f**) and the evening overpasses (**g**,**h**). The Sentinel-1 observations are shown separately for weak and strong winds (average wind speeds in the hour of the Sentinel-1 observation below and above 2.5 m s$^{-1}$, see the text in the sub-figures). Each point on the lines represents a bin of 25 Sentinel-1 observations along the x-axis. The Sentinel-1 observations masked for frozen conditions or rain interception are not included.

Both dew and intercepted rain wet the land surface. However, whereas the Sentinel-1 observations are affected by rain interception (see Section 4.2.3), they seem unaffected by dew. Gillespie et al. [28] and Riedel et al. [29] suggested that the nature of the intercepted water by dew and rain interception may be different (i.e., drop size or thin layer) and that this can alter their effect on the $\sigma^0$. Moreover, dew deposits are relatively small, typically 0.1 mm to 0.3 mm per night with maxima of 0.5 mm per night [27]. The rain interception storage capacity is generally larger, in the order of 2 mm for a Douglas-fir forest [69] and a dense mixed forest [24] in the Netherlands.

### 4.2.5. Summary of the Masking Rules

Tables 4 and 5 summarize the masking rules that result from the analyses in Sections 4.2.1–4.2.4 and list the mean effects on Sentinel-1 $d\sigma^0$ by the weather-related surface conditions.

The developed masking procedure for frozen conditions, snow, rain interception and dew is based on a rather simple set of masking rules and standard meteorological measurements, which adds to their applicability. However, in Sections 4.2.1–4.2.4 various complexities in the representations of these weather-related surface conditions and their effects on $\sigma^0$ were discussed. In general, the representations of the weather-related surface conditions could be improved by using models that simulate these surface conditions and by using available spatially distributed products from land surface models and satellite observations. For example, Kerr et al. [32] used the surface soil temperature and snow cover products from two models to mask SMOS observations, and Wagner et al. [31] used SSM/I satellite information about snow to flag ASCAT soil moisture retrievals. Even the surface state flags that accompany coarse resolution soil moisture products could be used as information in the masking of Sentinel-1 products.

It should be realized that due to uncertainty in the representations of the weather-related surface conditions and uncertainty in the SAR observations, the masks will remain imperfect. Some $\sigma^0$ observations will unnecessarily be masked (false alarms), whereas other observations will be disturbed but missed by the masking rules (missed hits). For example, not all Sentinel-1 observations that were acquired when air temperature was below $1\,^\circ\mathrm{C}$ are actually disturbed by frozen conditions, which is reflected in the dispersion of the points in Figure 5 and in Figure 4b for example in the period December 2016 to January 2017. This could be improved by utilizing the actual values of $\sigma^0$ observations [30], e.g., only masking for frozen conditions when the $\sigma^0$ or $d\sigma^0$ values of a Sentinel-1 observation indicate that it is affected. Further research is required to obtain the appropriate $\sigma^0$ or $d\sigma^0$ thresholds, thereby considering the $\sigma^0$ deviations that can be expected due to the $s_{S1}$ for a certain surface area (see Section 4.3).

**Table 4.** The masking rules for the weather-related surface conditions that result from the analyses in Sections 4.2.1–4.2.4. $T_{air}(t)$ is air temperature, $D_s(t)$ is snow depth and $P(t)$ is rainfall representative for the time of the Sentinel-1 observation ($t$). The calculation of these variables is explained in Section 3.2. The right columns list the mean $d\sigma^0$ of the Sentinel-1 VV observations to which a specific masking rule applies and none of the other masking rules apply.

| Surface Condition | Masking Rule | Mean of $d\sigma^0_{VV}$ [dB] | | |
|---|---|---|---|---|
| | | **Forests** | **Meadows** | **Cultivated Fields** |
| Frozen conditions | $T_{air}(t) \leq 1\,^\circ\mathrm{C}$ | −1.0 | −0.88 | −2.39 |
| Snow | $D_s(t) > 0\,\mathrm{cm}$ & $D_s$ at 9:00 CET after $t > 0\,\mathrm{cm}$ & Sentinel-1 morning observation & Land cover meadow or cultivated | Not masked | −0.52 | −1.93 |
| Rain interception | $\sum_{i=t-12\,hours}^{i=t} P(i) \geq 1.8\,\mathrm{mm}$ | 0.36 | 0.73 | 1.39 |
| Dew | No masking | Not masked | Not masked | Not masked |

**Table 5.** Same as Table 4, for the Sentinel-1 VH observations.

| Surface Condition | Masking Rule | Mean of $d\sigma^0_{VH}$ [dB] | | |
|---|---|---|---|---|
| | | **Forests** | **Meadows** | **Cultivated Fields** |
| Frozen conditions | $T_{air}(t) \leq 1\,°C$ | $-1.56$ | $-1.96$ | $-2.99$ |
| Snow | $D_s(t) > 0\,cm$ &<br>$D_s$ at 9:00 CET after $t > 0\,cm$ &<br>Sentinel-1 morning observation &<br>Land cover meadow or cultivated | Not masked | $-1.09$ | $-2.04$ |
| Rain interception | $\sum_{i=t-12\,hours}^{i=t} P(i) \geq 1.8\,mm$ | 0.43 | 0.91 | 1.53 |
| Dew | No masking | Not masked | Not masked | Not masked |

*4.3. Radiometric Uncertainty*

Figure 10 shows the $s_{S1}$, calculated with Equation (2), as a function of $A$ for the five selected forests and for each orbit in which Sentinel-1 collected data over the study region. As expected, the $s_{S1}$ improves for larger $A$ over which the Sentinel-1 $\sigma^0$ observations are averaged. The differences between the five forests in Figure 10 are rather small and there is no obvious systematic pattern between the forests, e.g., one forest consistenly showing the highest radiometric uncertainty in each sub-figure. This provides a justification for combined fits over the five forests, of which the model coefficients are listed in Tables 6 and 7.

The $s_{S1}$ is also rather similar for the different orbits, as can be seen in the blue and black lines in Figure 10. Only for the VV observations in orbit 15 there is a small underestimation of 0.03 dB at 10 ha by the combined fit, and for the VV observations in orbit 88 there is a small overestimation of 0.04 dB at 10 ha. These deviations are small and not seen in the other sub-figures, which suggests that there are no significant differences between the radiometric uncertainties of the different orbits due to the differences in incidence angle and pass direction (listed in Table 2). This is in accordance with the results by Schmidt et al. [9] and Schwerdt et al. [8], who draw similar conclusions in their analyses of the radiometric accuracy of the Sentinel-1A and Sentinel-1B observations acquired in the IW mode.

The combined fits over the four orbits and the five forests indicate that the $s_{S1}$ reduces from 0.85 dB (0.25 ha) to 0.30 dB (10 ha) for the VV polarization and from 0.89 dB (0.25 ha) to 0.36 dB (10 ha) for the VH polarization. Schwerdt et al. [8] also found a lower radiometric accuracy standard deviation for the Sentinel-1B VV observations (0.23 dB) than for the VH observations (0.33 dB). The developed masking rules have improved the $s_{S1}$ by up to 0.16 dB and 0.29 dB at 10 ha, for the VV and VH polarization respectively.

Interesting to note is that the $c_2$ coefficients in Tables 6 and 7 are close to −0.5, especially for the combined fits over the five forests and the four orbits. This value of −0.5 for $c_2$ is in accordance with the theory about the standard deviation of sample means (decreasing with $1/\sqrt{n}$), as explained in Section 3.3.

El Hajj et al. [10] obtained $s_{S1}$ values of 0.35 dB for VV and 0.45 dB for Sentinel-1 VH observations of a 8.1 ha forest area (for the longest period that El Hajj et al. [10] analysed). The $s_{S1}$ values calculated with the model coefficients in Tables 6 and 7 for $A$ equal to 8.1 ha are slightly lower (0.31 dB for VV and 0.37 dB for VH), which can be explained by the removal of the seasonal dynamics (see Section 4.1) and the extensive masking procedure in this study. The even lower $c_3$ coefficients in Tables 6 and 7 suggest that the $s_{S1}$ further decreases for $A$ larger than 10 ha. El Hajj et al. [10] also found lower $s_{S1}$ for $A$ larger than 10 ha, namely 0.19 dB (VV $\sigma^0$) for a racetrack of 14.7 ha, and 0.18 dB (VV $\sigma^0$) and 0.29 dB (VH $\sigma^0$) for a tropical forest of 635.2 ha. The $c_3$ values are also lower than the specified radiometric accuracy standard deviations of Sentinel-1A and Sentinel-1B (see Section 2.2). It should, however, be realized that the relationships of $s_{S1}$ versus $A$ have been developed on Sentinel-1 $\sigma^0$ observations aggregated over areas up to 10 ha. The extrapolation to larger spatial domains should be

tested on Sentinel-1 images collected over a study region with a time-invariant target of at least 100 ha (1 km × 1 km resolution, see Section 4.4).

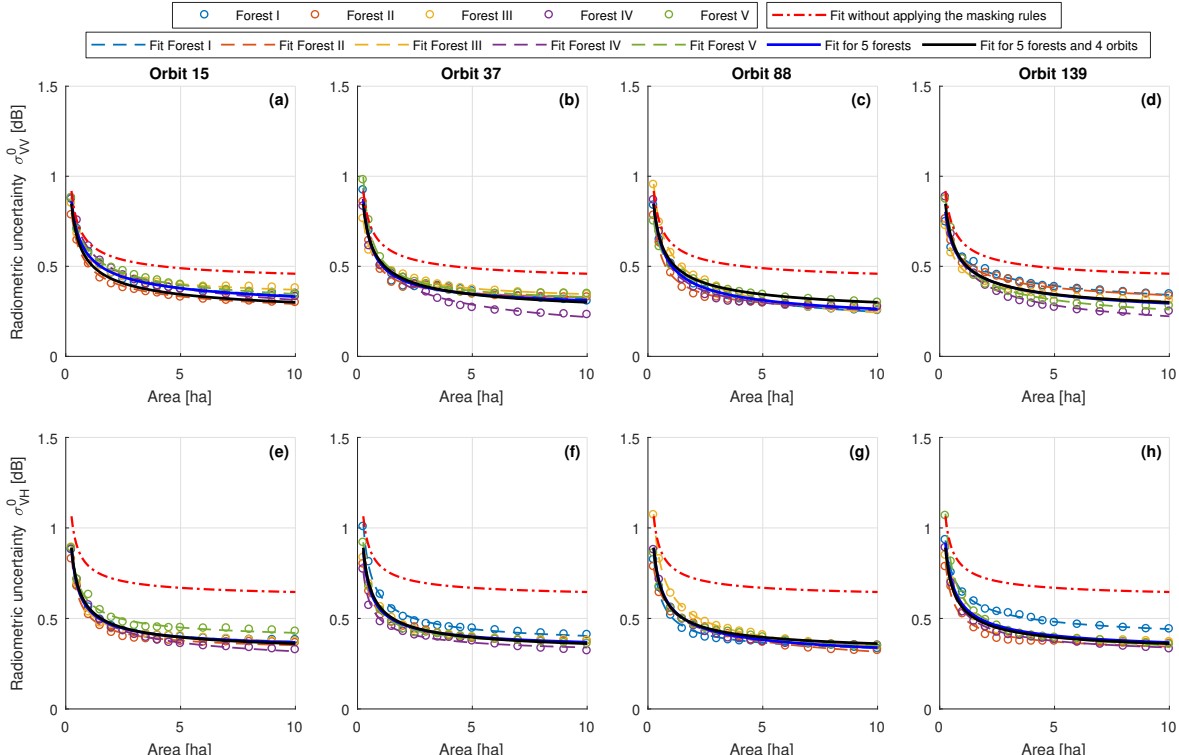

**Figure 10.** Radiometric uncertainty ($s_{S1}$) of the Sentinel-1 VV observations (**a**–**d**) and the Sentinel-1 VH observations (**e**–**h**). The points are calculated $s_{S1}$ values and the lines are the fitted second-order power functions (Equation (3)). The $s_{S1}$ is calculated on the forest $\sigma^0$ timeseries after application of the developed masking rules. For comparison the red lines show the combined fits over the four orbits and the five forests of $s_{S1}$ without applying the masking rules. The model coefficients of the fits are listed in Tables 6 and 7.

**Table 6.** Model coefficients of the second-order power function between $A$ and $s_{S1}$ (Equation (3)) of the Sentinel-1 VV observations over the five forests combined. The $E_{RMS}$ is the root mean squared deviation between the function and the underlying data points.

| Relative Orbit | $c_1$ | $c_2$ | $c_3$ | $E_{RMS}$ [dB] |
|---|---|---|---|---|
| All orbits in one fit | 0.3381 | -0.4809 | 0.1884 | 0.043 |
| 15 | 0.3556 | −0.4443 | 0.2042 | 0.029 |
| 37 | 0.2710 | −0.6091 | 0.2439 | 0.040 |
| 88 | 0.3706 | −0.4703 | 0.1377 | 0.031 |
| 139 | 0.3756 | −0.4035 | 0.1461 | 0.042 |

**Table 7.** Same as Table 6, for the $s_{S1}$ of the Sentinel-1 VH observations.

| Relative Orbit | $c_1$ | $c_2$ | $c_3$ | $E_{RMS}$ [dB] |
|---|---|---|---|---|
| All orbits in one fit | 0.2705 | -0.5765 | 0.2891 | 0.041 |
| 15 | 0.2376 | −0.6341 | 0.3125 | 0.030 |
| 37 | 0.2355 | −0.6228 | 0.3128 | 0.039 |
| 88 | 0.3314 | −0.4940 | 0.2325 | 0.037 |
| 139 | 0.2872 | −0.5636 | 0.2888 | 0.049 |

*4.4. Impact on Soil Moisture Retrievals*

Figure 11 shows the deviations in soil moisture retrievals that are expected due to the disturbing effect on the $\sigma^0$ by the weather-related surface conditions, if the Sentinel-1 observations would not be masked. The methodology explained in Section 3.4 was used to estimate the impacts of the mean effects on Sentinel-1 VV $\sigma^0$ observations over meadows and cultivated fields by frozen conditions, snow and rain interception (listed in Table 4) on soil moisture retrievals. Rain interception causes an overestimation, and snow and frozen conditions cause an underestimation of soil moisture. It should be noted that the decrease in soil moisture retrievals during frozen conditions relates to an actual decrease in liquid water content and this is, therefore, difficult to label as an error in soil moisture retrievals.

The penetration depth of C-band observations in wet snow is in the order of 3 cm to 1 m [6,70–72]. This suggests that for shallow snowpacks, like they occur in the study region, the effects of frozen conditions and wet snow may superimpose to some extent, decreasing the $\sigma^0$ further and causing even a larger underestimation of the soil moisture.

Figure 12 shows the expected soil moisture retrieval uncertainties due to the $s_{S1}$ as a function of $A$, for the surface conditions introduced in Section 3.4. The $s_{S1}$ is estimated with Equation (3) and the model coefficients listed in Table 6 for the combined fit over all orbits. With this, we assume that the $s_{S1}$ that has been estimated using forest $\sigma^0$ observations is applicable also to meadows and cultivated fields. This implicates that the the radiometric uncertainty sources are multiplicative, which is commonly adopted for speckle filtering [73–75].

The impacts on soil moisture retrievals are larger for the cultivated fields than for the meadows for two reasons. Firstly, regarding the impacts of the disturbing effects on the $\sigma^0$ by the weather-related surface conditions (Figure 11), the mean effects of frozen conditions, snow and rain interception on $\sigma^0$ are larger for the cultivated fields than for the meadows (see Table 4). Secondly, the $\sigma^0$ sensitivity to soil moisture is slightly larger for the surface roughness parameters taken as representative for meadows than for cultivated fields. As a result, a similar deviation in $\sigma^0$ results in a larger deviation in soil moisture for cultivated fields. However, as can be deduced from Figure 12, this effect is small in comparison to the first-mentioned reason.

The differences between the incidence angles of 35° and 44° are very small. In contrast to the small effects of surface roughness and incidence angle is the effect of soil moisture large. This is explained by the $\sigma^0$ sensitivity to soil moisture, which diminishes with increasing soil moisture according to IEM simulations [49]. This is also reflected in the larger values for $\Delta\theta^+$ in comparison to $\Delta\theta^-$ shown in Figure 12. Verhoest et al. [60] also showed soil moisture retrieval distributions that are skewed towards the higher soil moisture levels due to the non-linear relation between soil moisture and $\sigma^0$.

Paloscia et al. [61] proposed a spatial resolution of 1 km or finer for an operational Sentinel-1 based soil moisture product. Figure 13 shows the soil moisture retrieval uncertainty for the $s_{S1}$ that is estimated for a 1 km resolution (0.23 dB for the VV polarization), assuming that the obtained relations between $A$ and $s_{S1}$ can be applied to $A$ larger than 10 ha. The figure shows that the $s_{S1}$ consumes a significant portion of the 0.05 m$^3$ m$^{-3}$ accuracy requirement proposed for the operational soil moisture product by Paloscia et al. [61], especially in the wet soil moisture range.

With IEM we demonstrated the effect of various surface conditions on the soil moisture retrieval errors and uncertainty, for a surface without vegetation. Vegetation typically reduces the $\sigma^0$ sensitivity to soil moisture (e.g., [14]). As such, the impacts of the disturbing effects by the weather-related surface conditions and $s_{S1}$ on soil moisture retrievals are expected to increase with vegetation present. The impacts on soil moisture retrievals that we obtained with IEM, therefore, resemble the lower limits. The vegetation type and the vegetation stage can have a large effect on the $\sigma^0$ sensitivity to soil moisture, and further research is required to determine their effect on the soil moisture retrieval errors and uncertainty due to the weather-related surface conditions and the radiometric uncertainty.

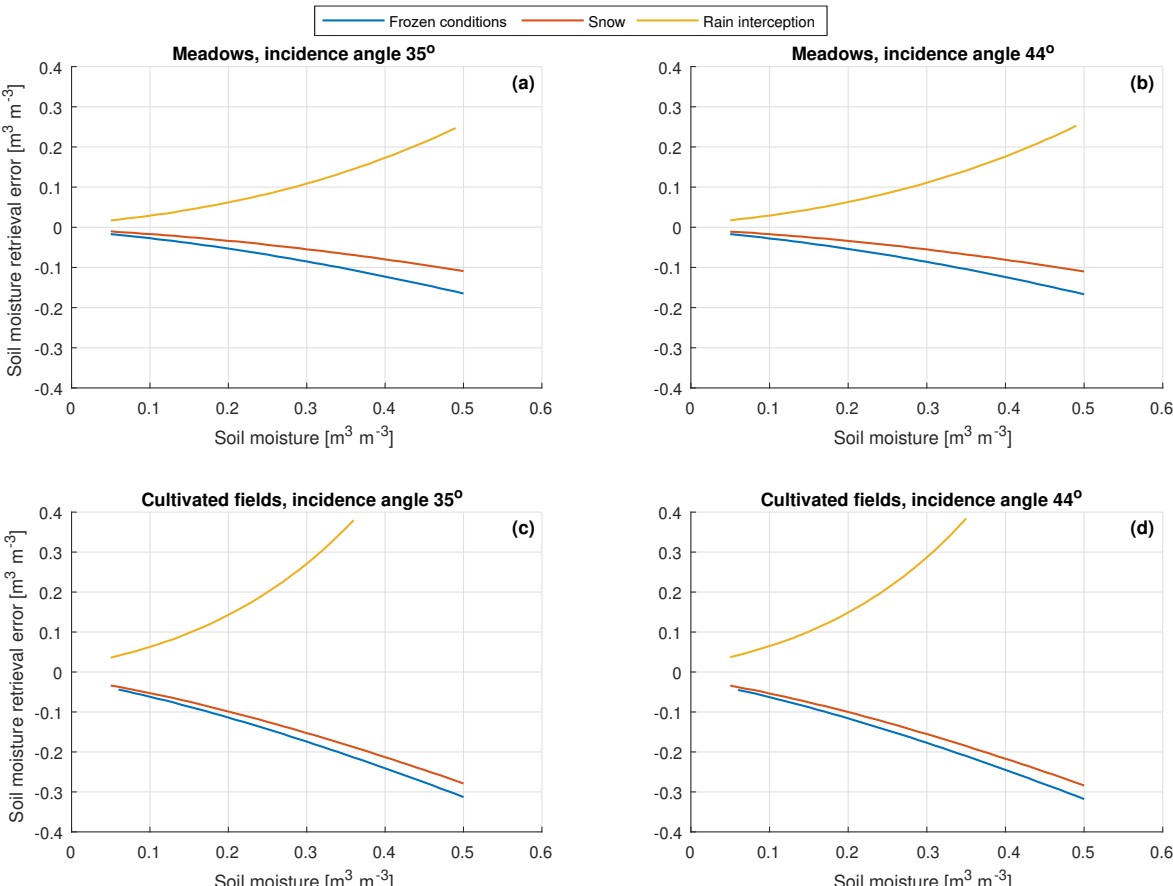

**Figure 11.** Impacts of frozen conditions, snow and rain interception on soil moisture retrieved from Sentinel-1 VV $\sigma^0$, for an incidence angle of 35° on meadows (**a**), 44° on meadows (**b**), 35° on cultivated fields (**c**) and 44° on cultivated fields (**d**). For the effects of frozen conditions, snow and rain interception on Sentinel-1 $\sigma^0$, the mean effects listed in Table 4 are taken.

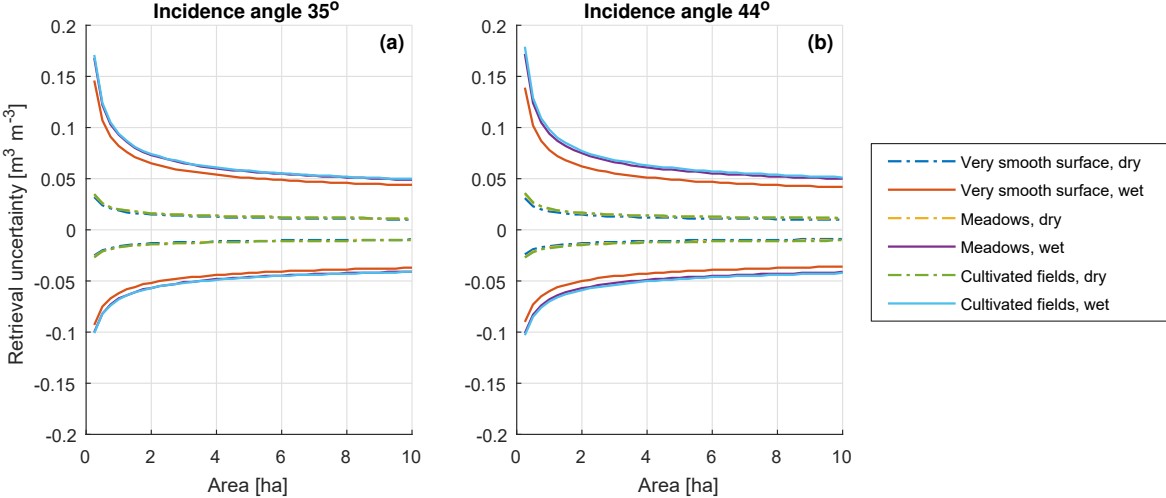

**Figure 12.** Upper ($\Delta\theta^+$) and lower boundary ($\Delta\theta^-$) of the soil moisture retrieval uncertainty due to the $s_{S1}$ at VV polarization as a function of $A$, for incidence angles of 35° (**a**) and 44° (**b**), for the three surface roughness scenarios (Table 3) and for dry soil (soil moisture equal to $0.10\,\mathrm{m}^3\,\mathrm{m}^{-3}$) and wet soil (soil moisture equal to $0.35\,\mathrm{m}^3\,\mathrm{m}^{-3}$).

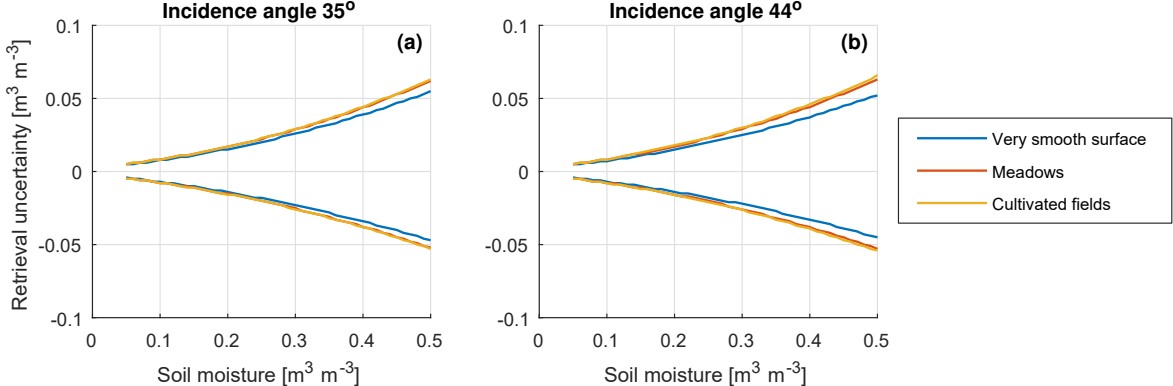

**Figure 13.** Upper ($\Delta\theta^+$) and lower boundary ($\Delta\theta^-$) of the soil moisture retrieval uncertainty due to the $s_{S1}$ at VV polarization at 1 km resolution as a function of soil moisture, for incidence angles of 35° (**a**) and 44° (**b**) and for the three surface roughness scenarios (Table 3).

## 5. Conclusions

In the scope of soil moisture retrieval from SAR observations, we have developed a masking procedure for weather-related surface conditions that disturb Sentinel-1 $\sigma^0$ observations and we have estimated the radiometric uncertainty of Sentinel-1 as a function of the surface area over which the $\sigma^0$ observations are aggregated. The impacts of the weather-related surface conditions and the radiometric uncertainty on soil moisture retrievals from Sentinel-1 $\sigma^0$ were investigated for various bare surface conditions using the IEM surface scattering model.

The effects of frozen conditions, snow, rain interception and dew on $\sigma^0$ were investigated by analysing meteorological measurements representing these surface conditions against seasonal anomalies of Sentinel-1 observations for five forests, five meadows and five cultivated fields in the eastern part of the Netherlands. From these analyses we have developed a set of masking rules for Sentinel-1 observations. From our results it follows that:

1. Sentinel-1 $\sigma^0$ observations of forests, meadows and cultivated fields are affected by frozen conditions below an air temperature of 1 °C, on average by −1.80 dB. Intercepted rain affects the $\sigma^0$ after more than 1.8 mm of rain in the 12 h preceding an observation, on average by +0.89 dB. Snow affects the $\sigma^0$ observations of meadows and cultivated fields that were acquired in the Sentinel-1 morning overpasses on average by −1.40 dB. We could not find a systematic effect of dew on the Sentinel-1 $\sigma^0$ observations, so no masking rule for dew was formulated.
2. Frozen conditions, snow and intercepted rain have the largest effect on the Sentinel-1 $\sigma^0$ observations of cultivated fields in comparison to forest and meadow $\sigma^0$ observations.
3. The effects of frozen conditions, snow and intercepted rain are larger on the Sentinel-1 $\sigma^0$ observations in VH than in VV polarization.

After masking the Sentinel-1 $\sigma^0$ timeseries of the five forests using the masking rules defined at point 1 above, Sentinel-1's radiometric uncertainty has been estimated by the standard deviation of the seasonal anomalies timeseries of the $\sigma^0$ observations averaged over forest surface areas ranging from 0.25 ha to 10 ha. From our results it follows that:

1. The Sentinel-1 radiometric uncertainty improves from 0.85 dB (0.25 ha) to 0.30 dB (10 ha) for the VV polarization and from 0.89 dB (0.25 ha) to 0.36 dB (10 ha) for the VH polarization.
2. The radiometric uncertainty is approximately inversely proportional to the square root of the surface area over which the Sentinel-1 $\sigma^0$ observations are averaged.

With the quantifications of radiometric uncertainty and the disturbing effects on the $\sigma^0$ by the weather-related surface conditions, we have determined their impact on soil moisture retrievals from Sentinel-1 VV $\sigma^0$. With the IEM model the $\sigma^0$ sensitivity to soil moisture was simulated for a surface

representing meadows and a surface representing cultivated fields, for the incidence angles at which Sentinel-1 observes the study region and for dry to wet soil conditions. Based on the results, we draw the following conclusions:

1.  If not masked, intercepted rain causes a significant overestimation of soil moisture ranging from $+0.047 \, \text{m}^3 \, \text{m}^{-3}$ for dry soils (soil moisture equal to $0.10 \, \text{m}^3 \, \text{m}^{-3}$) up to $+0.26 \, \text{m}^3 \, \text{m}^{-3}$ for wet soils (soil moisture equal to $0.35 \, \text{m}^3 \, \text{m}^{-3}$), averaged over the meadows and cultivated fields. Snow and frozen conditions lead to a significant decrease in soil moisture retrievals, from $-0.035 \, \text{m}^3 \, \text{m}^{-3}$ and $-0.045 \, \text{m}^3 \, \text{m}^{-3}$ for dry soils up to $-0.13 \, \text{m}^3 \, \text{m}^{-3}$ and $-0.16 \, \text{m}^3 \, \text{m}^{-3}$ for wet soils respectively.

2.  The soil moisture retrieval uncertainty as a result of radiometric uncertainty is minimum $-0.01 \, \text{m}^3 \, \text{m}^{-3}$ to $+0.01 \, \text{m}^3 \, \text{m}^{-3}$ for dry soils and large surface areas, and maximum $-0.10 \, \text{m}^3 \, \text{m}^{-3}$ to $+0.17 \, \text{m}^3 \, \text{m}^{-3}$ for wet soils and small surface areas.

3.  At the 1 km spatial resolution that Paloscia et al. [61] proposed for an operational Sentinel-1 soil moisture product, radiometric uncertainty still leads to soil moisture retrieval uncertainty ranging from $0.01 \, \text{m}^3 \, \text{m}^{-3}$ for dry soils to $0.033 \, \text{m}^3 \, \text{m}^{-3}$ for wet soils. Especially in the wet soil moisture range the radiometric uncertainty consumes a significant portion of the $0.05 \, \text{m}^3 \, \text{m}^{-3}$ accuracy requirement proposed for this soil moisture product [61].

4.  The impact on soil moisture retrievals by a $\sigma^0$ deviation, either due to a weather-related surface condition or radiometric uncertainty, is weakly dependent on the surface roughness and the incidence angle, and strongly dependent on the soil moisture itself.

This study demonstrates that the weather-related surface conditions disturbing Sentinel-1 $\sigma^0$ observations and Sentinel-1's radiometric uncertainty have a major impact on soil moisture retrieval uncertainty especially in wet soil conditions and for retrievals based on a small number of independent $\sigma^0$ samples (fine spatial resolutions). This understanding aids appreciating the application value of SAR based soil moisture products under various surface conditions and spatial resolutions. The reported uncertainty estimates represent the lower limits because the effects of vegetation are not accounted for in the simulations of $\sigma^0$ sensitivity to soil moisture. Further development of the masking procedure and characterization of other error contributions to soil moisture retrievals, such as imperfections in retrieval algorithms, would further benefit soil moisture product development and utilization.

**Supplementary Materials:** The following are available online at http://www.mdpi.com/2072-4292/11/17/2025/s1, Supplement 1: Sentinel-1 Backscatter timeseries of the Study Fields, Supplement 2: Processed Research Data Tables.

**Author Contributions:** Conceptualization, H.-J.F.B., R.v.d.V. and Z.S.; Data curation, H.-J.F.B.; Formal analysis, H.-J.F.B.; Funding acquisition, R.v.d.V. and Z.S.; Investigation, H.-J.F.B.; Methodology, H.-J.F.B. and R.v.d.V.; Project administration, R.v.d.V. and Z.S.; Resources, H.-J.F.B., R.v.d.V. and Z.S.; Software, H.-J.F.B.; Supervision, R.v.d.V. and Z.S.; Validation, R.v.d.V.; Visualization, H.-J.F.B.; Writing–original draft preparation, H.-J.F.B.; Writing–review and editing, H.-J.F.B., R.v.d.V. and Z.S.

**Funding:** This work is part of the research programme OWAS1S (Optimizing Water Availability with Sentinel-1 Satellites) with project number 13871, which is partly financed by the Dutch Research Council (NWO).

**Acknowledgments:** We thank all OWAS1S programme partners for their contributions. The Sentinel-1 images were downloaded from the Copernicus Open Access Hub (https://scihub.copernicus.eu/) [47]. The KNMI weather station and precipitation station data were obtained from http://www.knmi.nl/nederland-nu/klimatologie-metingen-en-waarnemingen [43].

**Conflicts of Interest:** The authors declare no conflict of interest. The funders had no role in the design of the study; in the collection, analyses, or interpretation of data; in the writing of the manuscript, or in the decision to publish the results.

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
