# Peer review of "Impacts of Radiometric Uncertainty and Weather-Related Surface Conditions on Soil Moisture Retrievals with Sentinel-1"

_remotesensing, doi:10.3390/rs11172025_

Round 1
Reviewer 1 Report
Manuscript Title:
Impacts of radiometric uncertainty and weather-related surface conditions on soil moisture retrievals with Sentinel-1
Major Comments:
1. The introduction section reads more like a mixture of literature review and methodology. Please separate the methods used in this study from the introduction and place them in the following method section. Please also reorganized the introduction as the following order: literature review, clearly present the research gap, and accordingly state the objectives of this study. And more focus will be given to past studies in the literature review.
2. The radiometric uncertainty was claimed to be estimated from forest stands in this study, and then was applied to meadow and cultivated sites, so as Eq. (3) describing the relationship between A and Ssi, the assumption behind which is that there is no difference in the radiometric uncertainties among forests and other two land cover types. Is there any reference support the point or any discussion on the uncertainties brought by this assumption? Also, tree species, or conifers or deciduous, were not introduced for the forest study sites.
3. The masking rules were listed in Tables 4 and 5 in the beginning of Section 4, however, it was not explicitly stated in the following sections that how these rules were determined based on the results presented in Figs. 5-9.
4. The masking of snow needs more explanations and discussion, i.e. Figs. 6 and 7. 1) Why forests seem not affected? The assumption in L328-239 is made purely out of free guess without the support from any reference; 2) the location of snow is not explicitly defined, i.e. if the c-band SAR signal is responding to the snow on the ground or in the canopy, and this depends on the canopy height which varies among different species. 3) I am not sure how the data points in Fig. 6 correspond to those in Fig. 7, there are positive y values for meadow in Fig. 7, are there any explanations? 4) Fig. 7, there is still negative effects from the snow on forests in this figure, is there really no need to apply masking rules for forests?
5. The masking threshold of 1.8mm for the 12-hour sum of rainfall should be treated more cautiously. This threshold may not be universal but depends on how much rainwater can be held by the tree canopy, depending on leaf amount and leaf structure.
6. For the effect of dew in Fig. 9, there is difference between the morning and evening passes and no explanation was given. Also, there is a negative effect of 1dB for the morning passes when RH= ~65%.
7. Since it is hard to distinguish the effect of snow and frost, there could be overlapping in the negative effects caused by them. In Fig. 11, the soil moisture error could be overestimated if we linearly add up the blue and red lines.
8. Figure captions need to be more specific, e.g. explain the meaning of every symbol used in the figures, even if the symbol has been explained in the text. Figure styles are not consistent, e.g. different styles between Figs. 6&7 and Figs. 5&8&9, which may be not a big deal as they are presenting the correct data but could be distracting for readers.
9. The manuscript needs to be thoroughly examined for grammar errors, such as incomplete sentences, the use of “the”, and etc.
10. In general, the masking rules summarized in this manuscript is very helpful for excluding low-quality Sentine-1 data that may affect the following retrieval of soil moisture. However, the manuscript is focused on testing the theoretical model sensitivity to the uncertainties/seasonal anomalies of Sentinel-1 data and could be strengthened by including the validation against ground-truth soil moisture measurements, which could be a strong prove of the power of the masking procedures. After thoroughly reviewing this manuscript, I therefore suggest a major revision.
Point-to-point Comments:
1. L36-37: please rephrase the sentence, especially please define “masking” under this context since this concept will be mentioned repeatedly in the text to come.
2. L40: -> disturbances in the…
3. L43-56: this is a description of the method, consider moving to the methodology section.
4. L48: -> After the application of …
5. L58-62: please rephrase
6. L60: increasing the window size of aggregation of pixels?
7. L65-67: I do not see the effect of speckle in this example.
8. L70: “to this aim”, please modify
9. L67-71: again, this fits better in the method section.
10. L73-74: the change of water level of leaves within the canopy may be another factor altering the backscattering of radar.
11. L75: “instabilities” due to what reasons?
12. L76-77: meaning not clear, please rephrase
13. L82-83: sentence not completed, please rephrase
14. L83-85: Need a summary of objectives of this study.
15. L89: -> “with measurements of soil moisture…”
16. L94: what kind of forest are these sites, deciduous, conifer or mixed? This should be stated since the two types of forests have distinctive tree and canopy structures, and thus should respond differently to microwave signals.
17. Fig. 1: reorganize the figure, e.g. to keep the same margins for all sub-figures on all sides; there are two (b) labels for the map and the legend, considering the combine the two.
18. L101: RNMI?
19. L126: over the 10 ha for the forest sites? What about meadow and cultivated sites?
20. L159-160: Thiessen polygons, or the so-called “Voronoi polygons” could be more appropriate to determine a station belongs to which study site, although this will lead to some sites have < 3 stations associated with them. This applies to the processing of other meteorological factors.
21. L193: A paragraph does not start from a symbol.
22. L249: “moving average dynamics” means? Do you mean the amplitude of the time series of VV polarization?
23. L252: What does “the minimum-to-maximum moving average amplitudes” mean? If these forested sites are comparable to the forests in Ref. #17 and #18 in terms of site conditions, e.g. tree species and tree density?
24. Tables 4 and 5, specify ‘-‘ means no obvious effects, and no need to perform the masking? More detailed figure captions for the explanation of the masking rules.
25. L276: “this is visible in Fig. 4”, please specify.
26. L278, specify the meaning of , even it was explained in L215.
27. L282, Any reasons why the decrease in d_sigma_naught with Tair is more obvious in VH and in cultivated fields? Can this be explained in terms of the seasonality of vegetations over three types of land cover?
28. Fig. 5, What about the decrease for the cultivated fields at about 28 degree C?
29. L298: factor 6?
30. L291-305, the explanation on Fig. 6 is not complete, e.g. the difference among species
31. L319-324, if I understand correctly, the authors mean that during an evening overpass, there is no effect of snow because the snow has melted already during the day with Tair>0 degree C. My doubt is if this happening to a thicker snowpack, the melting may not complete in one day, thus the wet melting snow may decrease the SAR signals. Can this explain some low y values when snow depth is high in Fig. 6?
32. L348, can the ponding of water explain the drop at 4mm accumulated rainfall in Fig. 8?
33. L378, does this means that the effect of dew is negligible because the amount of water retained in the canopy by dew deposition is much smaller than that by a rainfall?
34. Fig. 10, a bit redundant and can be simplified, e.g. the “fit for 5 forests and 4 orbits” are shown for multiple times. Show one figure for each polarization and present the parameters in the figures.
35. L387, reference?
36. L387, -> “which states”
37. Fig. 12, modify the figure caption by adding the “area”.
38. L411-412, they look very similar to me in Fig. 12.
39. L468-270: rephrase the sentence.
40.
41. L14-15: The improvement is from masking or the aggregation or both? This is unclear.
42. L91: slope information for these area?
43. L94: do you have LAI measurement (or from MODIS) for the forest?
44. L136: should W is half of a window size?
45. Table 2: how the incidence angle affect the sigma_zero and then affect your analysis?
46. Section 3.2.2: wet snow and dry snow are quite different to affect the backscatter, do you have such information?
47. Table 3: need to indicate the meaning of each line/color
48. L207: Add some latest applications of IEM / AIEM model, such as Guo et al., “An Improved Approach for Soil Moisture Estimation in Gully Fields of the Loess Plateau Using Sentinel-1A Radar Images”, RS 2019, and He et al., “Simulation and SMAP Observation of Sun-Glint Over the Land Surface at the L-Band”, 2017, IEEE TGRS.
49. Figure 4: some of these masks are very effective; why some masked values are very close to the moving average, need some explanation. For frost in Jan 2017, it is clear to see signal drop for orbit 88 and 15, but for orbit 139, there is no much change, any reason?
50. Section 4.2.1 and fig. 5: some misleading here. This figure general indicates the freeze/ thaw status of canopy and ground?
51. L295: thanks. I see the discussion dry/wet snow here.
52. Figure 10 makes sense but it assumes that the ground land cover is very uniform. Even there is no noise (spike) in the backscatter, you can still get this pattern with aggregation of pixels. Can you add some justification about the homogeneity of your study area?
Section 4.4: the IEM model explains the signal from bare soil. Here you superpose all the uncertainty (due to rain/ snow …) to assuming a bare ground? I would expect an IEM + watercloud model here. But the description of method is not clear to me.
53. L442: that is the weakness of this study: you have done nice analysis due to these disturbance factors roughly from the canopy. But the impact on soil moisture is done assuming a bare soil, the transmittance effect is ignored.
54. L449: frost is the ice on leaf surface but here it is mixed with the condition of freeze of leaf and branch (inside of plant organ)?
55. L458: it is desirable to see another curve: how the “uncertainty” decreases without masking with the increase of surface area.
Reviewer 2 Report
This paper is well structured and well written. However, I recommend the authors to also validate the Sentinel soil moisture retrievals against insitu observations, which is a necessary sep.
L32 I found it strange to refer it to "radiometric uncertainty" while you are speaking of radars?
Figure 5: add in the caption what is a and b.
L192 What I see in Figure 6 is the mask of rain interception not snow depth?
L207 please add a description or a diagram of this model "IEM surface scattering model"
Figure 11: How "Soil moisture retrieval error" was calculated. Overestimation or underestimation based on what?please elaborate more here.
Reviewer 3 Report
I have read with interest the submitted manuscript entitled "Impacts of Radiometric Uncertainty and Weather-Reltaed Surface Conditions on Soil Moisture Retrievals with Sentinel-1". The authors present a very detailed analysis of the causes of wrong estimation of soil moisture with Sentinel-1 due to weather conditions (rain interception, dew, frost, snow) and radiometric uncertainty, that leads to quantitative conclusions in terms of accuracy and propose rules to mask out undesirable estimates.
I would like to congratulate the authors for this expected study and for providing a clear and nice-to-read manuscript, with full of information about the methodology and well-balanced in regard with illustrations and comments on the results. This study is very interesting, as many influencing factors have been scanned, with a strong validation material (15 sites), and as practical selection algorithm is proposed to anyone using such kind of data. Illustrations are clear, text well structured, and conclusions well drawn.
Only a very minor concern about dew effect: from Fig 9 a) and e), it seems there is a small effect by low wind speed and 98% humidity for cultivated fields (~1dB). It is said that there is no effect, but apparently there is in some cases (probably there is a possibility to refine the analysis and detail when there is dew affecting the signal...), so maybe a less strong statement would be necessary in the conclusion (and show a possibility to refine in the text). Nevertheless, the argumentation given is convincing enough to explain why it is different from rain interception.
Reviewer 4 Report
Overall well written and detailed paper. I do not have any specific comments.
Round 2
Reviewer 1 Report
Thanks to the authors for their excellent responses to my previous comments. This is a very important study.
Please examine the following in your proof:
L392: check "That this effect by frozen conditions ..."
L746: check if it is available online: "The following are available online at https://www.mdpi.com/2072-4292/xx/1/5/s1"